# ActionPiece: Contextually Tokenizing Action Sequences for Generative Recommendation

**Yupeng Hou** [1] [*]  **Jianmo Ni** [2]  **Zhankui He** [2]  **Noveen Sachdeva** [2]  **Wang-Cheng Kang** [2]
**Ed H. Chi** [2]  **Julian McAuley** [1]  **Derek Zhiyuan Cheng** [2]

## Abstract

Generative recommendation (GR) is an emerging paradigm where user actions are tokenized into discrete token patterns and autoregressively generated as predictions. However, existing GR models tokenize each action independently, assigning the same fixed tokens to identical actions across all sequences without considering contextual relationships. This lack of context-awareness can lead to suboptimal performance, as the same action may hold different meanings depending on its surrounding context. To address this issue, we propose ActionPiece to explicitly incorporate context when tokenizing action sequences. In ActionPiece, each action is represented as a *set* of item features. Given the action sequence corpora, we construct the vocabulary by merging feature patterns as new tokens, based on their co-occurrence frequency both within individual sets and across adjacent sets. Considering the unordered nature of feature sets, we further introduce set permutation regularization, which produces multiple segmentations of action sequences with the same semantics. Our code is available at: `https://github.com/google-deepmind/action_piece`.

## 1. Introduction

Generative recommendation (GR) (Tay et al., 2022; Rajput et al., 2023; Zheng et al., 2024; Zhai et al., 2024) is an emerging paradigm for the sequential recommendation task (Hidasi et al., 2016; Kang & McAuley, 2018). By tokenizing the user actions (typically represented by the

---
[*]Work done as a student researcher at Google DeepMind. [1]University of California, San Diego [2]Google DeepMind. Correspondence to: Yupeng Hou and Jianmo Ni <yphou@ucsd.edu, jianmon@google.com>.

*Proceedings of the 42nd International Conference on Machine Learning*, Vancouver, Canada. PMLR 267, 2025. Copyright 2025 by the author(s).

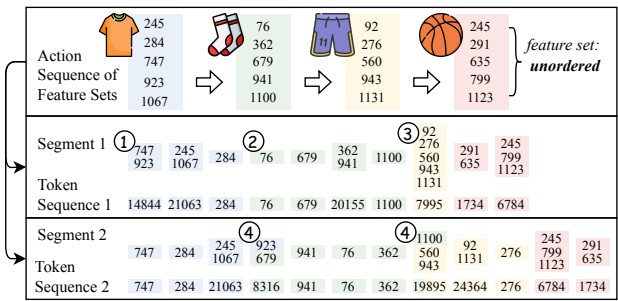

*Figure 1.* Illustration of the tokenization process of ActionPiece. Each action is represented as an unordered feature set. This figure presents two possible tokenized sequences, where features are grouped into different segments. The same action can be tokenized into different tokens depending on the surrounding context. A detailed case study can be found in Section 4.5.

interacted items) into discrete tokens, GR models learn to autoregressively generate tokens, which are then parsed into recommended items. These tokens share a compact vocabulary that does not scale with the item pool size, improving model scalability, memory efficiency, and recommendation performance. The input action sequence is vital in understanding user intentions (Hidasi et al., 2016; Li et al., 2017; Kang & McAuley, 2018), which organizes a user's historical interactions in chronological order. The same action (*e.g.,* purchasing the same item) may have different meanings in different action sequences. Evidence of taking a certain action can be found in the context, such as whether other items in the sequence share the same brand, color tone, or price range (Zhang et al., 2019; Zhou et al., 2020; Hou et al., 2022; 2023; Yuan et al., 2023).

Despite the importance of contextual relations among actions, existing methods tokenize each action independently of its context (summarized in Table 1). The typical pipeline for tokenizing action sequences involves two steps: (1) Tokenizing each action/item individually into a pattern of tokens; (2) Replacing each action in the input sequence with its corresponding token pattern. In this way, the tokens do not explicitly contain the context. Instead, they solely rely on the autoregressive model's parameters being well-trained to generalize effectively in understanding the context, which challenges the capabilities of GR models. As a compari-

son, tokenization in language modeling also originates from context-independent methods, such as word-level tokenization (Sutskever et al., 2014; Bahdanau et al., 2015). A decade of progress has led to most tokenization methods for modern large language models (LLMs) (OpenAI, 2022; Anil et al., 2023; Touvron et al., 2023; Zhao et al., 2023) adopting context-aware approaches, including BPE (Sennrich et al., 2016) and Unigram tokenization (Kudo, 2018), which tokenize the same word roots along with their adjacent context into different tokens.

In this work, we aim to make the first step towards context-aware tokenization for modeling *action sequences*. In analogy to how characters or bytes serve as the basic units in language modeling, we consider the associated features of an item as initial tokens. The idea is to iteratively find the most commonly co-occurring pairs of tokens among the training action sequences, then merge them into new tokens to represent segments of context. However, it's non-trivial to achieve this. Unlike text, where characters naturally form a sequence, the features associated with an action form an unordered set (Zhang et al., 2019; Zhou et al., 2020). Thus, the proposed tokenization algorithm should be applied on *sequences of token sets*. We need to carefully consider which pairs of tokens should be counted, whether within a single set or between two adjacent sets, and how much weight should be given to these different types of relationships.

To this end, we propose **ActionPiece**, which enables the same actions to be tokenized into different tokens based on their surrounding context. **(1) Vocabulary construction** begins by initializing the vocabulary to include every unique feature as initial tokens. The vocabulary is then constructed by iteratively learning merge rules. Each merge rule specifies that a pair of tokens can be merged into a new token. In each iteration, we enumerate the training corpus to count the co-occurrence of existing tokens. Considering the structural differences between token pairs, *e.g.,* whether they occur within a single set or between two adjacent sets, we assign different weights to different pairs. **(2) Segmentation** refers to dividing raw features in action sequences into groups that can be replaced by tokens from the vocabulary. To fully exploit the unordered nature of the feature set of each action, we introduce set permutation regularization. By randomly permuting the features within each set, we can produce multiple token sequences of a single action sequence that preserve the same semantics. These variations act as natural augmentations for training data and enable inherent ensembling during model inference.

## 2. Related Work

**Generative recommendation.** Conventional sequential recommendation models often relies on large embedding tables to store representations for all items, leading to signif-

*Table 1.* Comparison of different action tokenization methods for generative recommendation. "Contextual" denotes whether the same actions can be tokenized into different tokens based on the surrounding context. "Unordered" denotes whether the item features or semantic IDs are used in an order-agnostic manner.

| Action Tokenization | Example | Contextual | Unordered |
|---|---|---|---|
| Product Quantization | VQ-Rec (Hou et al., 2023) | ✗ | ✔ |
| Hierarchical Clustering | P5-CID (Hua et al., 2023) | ✗ | ✗ |
| Residual Quantization | TIGER (Rajput et al., 2023) | ✗ | ✗ |
| Text Tokenization | LMIndexer (Jin et al., 2024) | ✗ | ✗ |
| Raw Features | HSTU (Zhai et al., 2024) | ✗ | ✗ |
| SentencePiece | SPM-SID (Singh et al., 2024) | ✗ | ✗ |
| ActionPiece | Ours | ✔ | ✔ |

icant engineering and optimization challenges (Hidasi et al., 2016; Li et al., 2017; Kang & McAuley, 2018). Generative recommendation (Rajput et al., 2023; Zheng et al., 2024; Zhai et al., 2024; Deldjoo et al., 2024; Hou et al., 2025) addresses these issues by tokenizing each item as tokens from a shared vocabulary. By autoregressively generating the next tokens as recommendations, this generative paradigm offers benefits such as memory efficiency (Rajput et al., 2023; Yang et al., 2024; Ding et al., 2024), scalability (Zhai et al., 2024; Liu et al., 2024b), and easier alignment with LLMs (Zheng et al., 2024; Jin et al., 2024; Tan et al., 2024; Li et al., 2025). Existing research has developed different action tokenization techniques, such as hierarchical clustering (Hua et al., 2023; Si et al., 2024), quantization (Rajput et al., 2023; Wang et al., 2024a; Zhu et al., 2024a), or jointly training with recommendation models (Liu et al., 2025). Other works incorporate additional modalities like collaborative filtering (Petrov & Macdonald, 2023; Wang et al., 2024c;b; Liu et al., 2024b;a) and natural language (Zheng et al., 2024; Jin et al., 2024; Hou et al., 2024b; Zhang et al., 2025a). However, current methods tokenize each action independently, ignoring the surrounding context. In this work, we propose the first context-aware action tokenization method, where the same actions are tokenized differently in different action sequences.

**Tokenization for language modeling.** Tokenization is the process of transforming raw text into discrete token sequences (Kudo & Richardson, 2018). Early word-level methods are context-independent and struggle to tokenize out-of-vocabulary words (Sutskever et al., 2014; Bahdanau et al., 2015). Consequently, subword-level tokenization has gradually become the more mainstream choice. The vocabularies of these subword-level tokenizers are constructed iteratively, either bottom-up (starting with a small vocabulary and merging commonly occurring token pairs as new tokens) (Wu et al., 2016; Sennrich et al., 2016), or top-down (starting with a large vocabulary and pruning tokens to minimize likelihood decrease) (Kudo, 2018; Yehezkel & Pinter, 2023). Once the vocabulary is built, the text can be seg-

---

**Algorithm 1** ActionPiece Vocabulary Construction

---

**input** Sequence corpus $\mathcal{S}'$, initial tokens $\mathcal{V}_0$, target size $Q$
**output** Merge rules $\mathcal{R}$, constructed vocabulary $\mathcal{V}$

1: Initialize vocabulary $\mathcal{V} \leftarrow \mathcal{V}_0$ # Each initial token corresponds to one unique item feature
2: $\mathcal{R} \leftarrow \emptyset$
3: **while** $|\mathcal{V}| < Q$ **do**
4:     # *Count:* accumulate weighted token co-occurrences
5:     $\text{count}(\cdot, \cdot) \leftarrow \text{Count}(\mathcal{S}', \mathcal{V})$ # Algorithm 2
6:     # *Update:* merge a frequent token pair into a new token
7:     Select $(c_u, c_v) \leftarrow \arg\max_{(c_i, c_j)} \text{count}(c_i, c_j)$
8:     $\mathcal{S}' \leftarrow \text{Update}(\mathcal{S}', \{(c_u, c_v) \rightarrow c_{\text{new}}\})$ # Algorithm 3
9:     $\mathcal{R} \leftarrow \mathcal{R} \cup \{(c_u, c_v) \rightarrow c_{\text{new}}\}$ # New merge rule
10:    $\mathcal{V} \leftarrow \mathcal{V} \cup \{c_{\text{new}}\}$ # Add new token to the vocabulary
11: **end while**
**return** $\mathcal{R}, \mathcal{V}$

---

mented either using the same method employed during vocabulary construction or based on additional objectives (He et al., 2020; Provilkov et al., 2020; Hofmann et al., 2022; Schmidt et al., 2024). As an analogy, existing action tokenizers are context-independent and function like "word-level" language tokenizers. In this work, we take the first step toward context-aware subaction-level action tokenizer.

# 3. Method

In this section, we present **ActionPiece**, a context-aware method for tokenizing action sequences for generative recommendation. First, we formulate the task in Section 3.1. Then, we introduce the proposed tokenizer, covering vocabulary construction and segmentation, in Section 3.2. Finally, we describe the model training and inference process using ActionPiece-tokenized sequences in Section 3.3.

## 3.1. Problem Formulation

Given a user's historical actions $S = \{i_1, i_2, \dots, i_t\}$, organized sequentially by their timestamps, the task is to predict the next item $i_{t+1}$ the user will interact with.

**Action as an unordered feature set.** In the development of modern recommender systems, each item $i_j$ is usually associated with a set of features $\mathcal{A}_j$ (Zhang et al., 2019; Zhou et al., 2020; Cheng et al., 2016). Assuming there are $m$ features per item, the $k$-th feature of item $i_j$ is denoted as $f_{j,k} \in \mathcal{F}_k$, where $\mathcal{F}_k$ is the collection of all possible choices for the $k$-th feature. Compared to representing actions using ordered semantic IDs (*e.g.,* those produced by RQ-VAE (Rajput et al., 2023; Singh et al., 2024)), the unordered set setting offers two key advantages: (1) It does not require a specific order among features, which aligns better with how items or actions are represented in most recommender systems; (2) It enables the inclusion of more general discrete and numeric features, such as *category*, *brand*, and *price* (Pazzani & Billsus, 2007; Juan et al., 2016).

**Action sequence as a sequence of sets.** Representing each item as an unordered set, the input action sequence can be written as $S' = \{\mathcal{A}_1, \mathcal{A}_2, \dots, \mathcal{A}_t\}$, which is a chronologically ordered sequence of sets. There is no order within each set, but there are orders between the features from different sets. The tokenizer design should account for the ordered and unordered relationships among features.

**Generative recommendation task.** In this work, we aim to design a tokenizer that maps an input action sequence $S'$ to a token sequence $C = \{c_1, c_2, \dots, c_l\}$, where $l$ denotes the number of tokens in the sequence. Note that $l$ is typically greater than the number of actions $t$. Next, we train a GR model to autoregressively generate tokens $\{c_{l+1}, \dots, c_q\}$, which can be parsed as next-item predictions $\hat{i}_{t+1}$.

## 3.2. Contextual Action Sequence Tokenizer

The proposed tokenizer is designed to transform action sequences (represented as sequences of feature sets) into token sequences. In the ActionPiece-tokenized sequences, each token corresponds to a set containing varying numbers of features. For example, a token can represent: (1) a subset of features from one item; (2) a single feature; (3) all features of one item; or (4) features from multiple items. We also label these four types of tokens in Figure 1. Below, we first describe how to construct the ActionPiece tokenizer's vocabulary given a corpus of action sequences (Section 3.2.1). Then, we introduce how to segment action sequences into a new sequence of sets, where each set corresponds to a token from the constructed vocabulary (Section 3.2.2).

### 3.2.1. VOCABULARY CONSTRUCTION ON ACTION SEQUENCE CORPUS

Given a corpus of action sequences $\mathcal{S}'$, the goal of vocabulary construction is to create a vocabulary $\mathcal{V}$ of $Q$ tokens. Each token represents a combination of features that frequently occur in the corpus. Similar to BPE (Sennrich et al., 2016), we construct the vocabulary using a bottom-up approach. The process starts with an **initial vocabulary** of tokens $\mathcal{V}_0$. The construction proceeds iteratively, adding one new token to the vocabulary at each iteration until the predefined target size is reached. Each iteration consists of two consecutive steps: **count**, where the most frequently occurring token pair is identified, and **update**, where the corpus is modified by merging the selected pair into a new token. An algorithmic workflow is illustrated in Algorithm 1.

**Vocabulary initialization.** In BPE, each token represents a sequence of bytes. Thus, the most fundamental units–the initial tokens–are single bytes, which form the initial vocabulary of BPE. Similarly, each token in ActionPiece represents a set of features. Therefore, we initialize ActionPiece with a vocabulary in which each token represents a set containing one unique item feature. Formally, we denote the

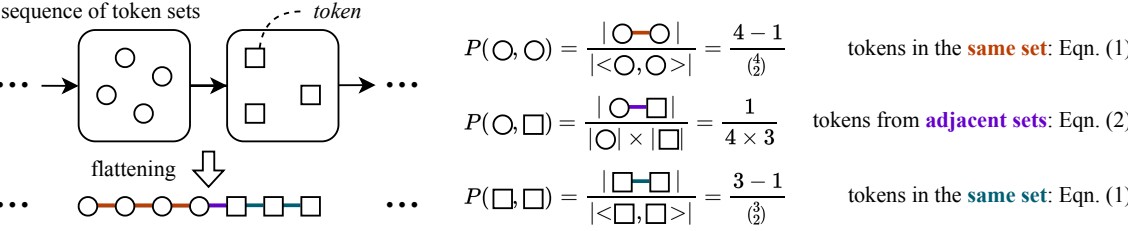

**Figure 2.** Illustration of how weights of co-occurring token pairs are counted during vocabulary construction. In this example, two adjacent sets in the sequence are considered: one with 4 tokens (represented as ◯) and another with 3 tokens (represented as ▢). Token pairs are counted within a single set ($< ◯, ◯ >$ and $< ▢, ▢ >$) and across the two adjacent sets ($< ◯, ▢ >$).

initial vocabulary as $\mathcal{V}_0 = \{c = \{f\} | f \in \mathcal{F}_1 \cup \ldots \cup \mathcal{F}_m\}$. After initializing the vocabulary, each action sequence (of feature sets) can be represented as a sequence of token sets.

**Count: context-aware token co-occurrence counting.** In each iteration of vocabulary construction, the first step is to count the co-occurrence of token pairs in the corpus. These pairs capture important feature combinations, which are encoded by creating new tokens. There are two types of token co-occurrence within a sequence of sets: (1) two tokens exist within the same set, or (2) two tokens exist in adjacent sets in the sequence. Notably, the second type allows ActionPiece to explicitly include context information.

*Weighted co-occurrence counting.* In one-dimensional token sequences (*e.g.,* text), all token pairs are typically treated equally. However, in sequences of token sets, token pairs vary based on their types and the sizes of their respective sets. To account for these differences, we propose assigning different weights to token pairs. To determine the weight for each token pair, we relate sequences of token sets to token sequences by randomly permuting the tokens within each set and flattening them into a single token sequence. Let $P(c, c')$ represent the expected probability that tokens $c$ and $c'$ are adjacent in the flattened sequence. For two tokens from the same set, we have:

$$P(c_1, c_2) = P(c_2, c_1) = \frac{|\mathcal{A}_i| - 1}{\binom{|\mathcal{A}_i|}{2}} = \frac{2}{|\mathcal{A}_i|}, \quad c_1, c_2 \in \mathcal{A}_i, \tag{1}$$

and for two tokens from adjacent sets, we have:

$$P(c_1, c_3) = \frac{1}{|\mathcal{A}_i| \times |\mathcal{A}_{i+1}|}, \quad c_1 \in \mathcal{A}_i, \ c_3 \in \mathcal{A}_{i+1}. \tag{2}$$

By considering the probabilities of all adjacent token pairs in the flattened sequence as 1, the weights for token pairs in the original sequence of token sets correspond to the probabilities given in Equations (1) and (2). An illustration is shown in Figure 2.

*Accumulating co-occurrence weights.* The weights described above are calculated based solely on the co-occurrence type and the set size. They do not take into account the specific tokens being analyzed. Tokens $c_i$ and

$c_j$ might appear in the same set in one sequence but in two adjacent sets in another sequence. By iterating through the corpus, we sum up the weights for each token pair whenever they appear together multiple times.

**Update: corpus updating with action-intermediate nodes.** The next step in each iteration is to merge the token pair with the highest accumulated co-occurrence weight. Since token merging may change the set size, we use a double-ended linked list (Zouhar et al., 2023) to maintain each action sequence, where each node represents a set of tokens. Merging tokens within the same set is straightforward, *i.e.,* replacing the two tokens with a new one. However, merging tokens from two adjacent sets is more complex, *e.g.,* determining which set should include the new token.

*Intermediate Node.* We introduce the concept of "intermediate node" to handle tokens that combine features from multiple sets. Initially, all nodes in the maintained linked lists contain features specific to their corresponding actions. These nodes are referred to as "action nodes."

(1) When tokens from two adjacent action nodes are being merged, we insert a new intermediate node between the two action nodes. The new token is stored in the intermediate node, and the merged tokens are removed from their respective action nodes;

(2) When merging tokens from an action node and an intermediate node, the new token replaces the original token in the intermediate node. The reason is that this new token also combines features from multiple actions. After the merge, the token from the action node is removed.

Following the above update rules ensures that there is at most one intermediate node between any two action nodes, and each intermediate node contains no more than one token. When calculating co-occurrence weights involving an intermediate node, it can simply be treated as a set of size 1.

**Efficient implementation.** Naively counting and updating the corpus requires a total time complexity of $O(QNLm^2)$, where $Q$ is the target vocabulary size, $N$ is the number of action sequences in the training corpus, and $L$ is the average length of these sequences. However, it is unnecessary to count co-occurrences from scratch in each iteration. This is

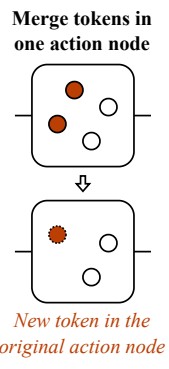
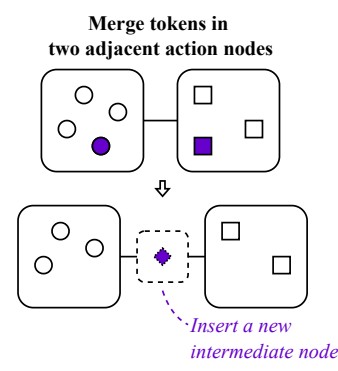
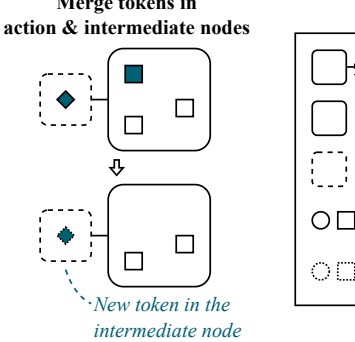
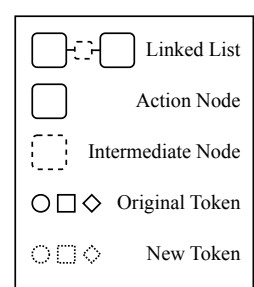

*Figure 3.* Illustration of how the linked list, which maintains the action sequence, is updated when merging two tokens into a new token. Three cases are considered: (1) both tokens are in the same action node; (2) the tokens are in two adjacent action nodes; (3) one token is in an action node, while the other is in an intermediate node.

because only a small portion of the maintained linked lists is modified compared to the previous iteration.

*Data structures.* To address this, we propose creating inverted indices to map token pairs to all the linked lists that contain them. A global heap is maintained to return the token pair with the highest accumulated co-occurrence. The key challenge lies in updating these data structures. We carefully compute the changes in accumulated co-occurrences and update the inverted indices. For the heap, we employ a lazy-update strategy. We insert the latest weights with a tag. When fetching a value from the heap, we check the tag to verify if the value is up-to-date. If it is not, we discard the value and fetch the next one.

*Time complexity.* Let $H = O(NLm)$ represent the maximal heap size. Using the proposed algorithm, we successfully reduce the original time complexity to $O(\log Q \log H \cdot NLm^2)$, achieving efficient vocabulary construction. In practice, the later iterations take significantly less time than the initial ones. This is expected and because tokens with higher accumulated co-occurrence weights typically appear frequently in the early stages. However, the overall construction time benefits from the reduced amortized complexity. Further details about the vocabulary construction algorithm are provided in Appendix C.

### 3.2.2. SEGMENTATION BY SET PERMUTATION REGULARIZATION

Segmentation is to convert original action sequences into a sequence of feature sets. Each set in the segmented sequence corresponds to a token in the vocabulary.

**Naive segmentation.** One segmentation strategy in ActionPiece involves applying the same technique used to construct the vocabulary. Specifically, this technique iteratively identifies token pairs with high priorities (represented by the IDs of tokens, where tokens added earlier may have higher priority). However, we observed that this strategy can lead

to a bias, where only a subset of tokens in the vocabulary is frequently used (as shown empirically in Section 4.4.2).

**Set permutation regularization (SPR).** To address this issue and account for the unordered nature of sets, we propose *set permutation regularization*, which generates multiple segmentations for each action sequence. The key idea is to avoid enumerating all possible pairs between tokens in a set or adjacent sets. Instead, we generate a random permutation of each set and treat it as a one-dimensional sequence. By concatenating all the permutations, we create a long token sequence. This sequence can then be segmented using traditional BPE segmentation methods (Sennrich et al., 2016). In this approach, different permutations can produce distinct segmented token sequences with the same semantics. These sequences serve as natural augmentations for model training (Section 3.3.1) and enable inherent ensembling during model inference (Section 3.3.2).

### 3.3. Generative Recommendation Models

#### 3.3.1. TRAINING ON AUGMENTED TOKEN SEQUENCES

For an action sequence and its ground-truth next action in the training corpus, we tokenize them into token sequences $C_{in}$ and $C_{out}$, respectively. Taking $C_{in}$ as input, we train a Transformer encoder-decoder module (Raffel et al., 2020) to autoregressively generate $C_{out}$ (*e.g.,* next-token prediction objective (Rajput et al., 2023)). During training, we tokenize the action sequence using the set permutation regularization described in Section 3.2.2 in each epoch. This approach naturally augments the training sequences, which empirically improves model performance, as shown in Section 4.3.

#### 3.3.2. INFERENCE-TIME ENSEMBLING

During model inference, we tokenize each action sequence $q$ times using set permutation regularization. By passing these $q$ tokenized sequences through the model, we obtain $q$ output ranking lists (*e.g.,* using beam search for inference

*Table 2.* Comparison between the widely used text tokenization method, byte-pair encoding (BPE) (Sennrich et al., 2016), and the proposed context-aware action sequence tokenization method, ActionPiece.

| Aspect | BPE | ActionPiece |
|---|---|---|
| **Data Type** | text sequences | action (unordered feature set) sequences |
| **Token** | a byte sequence | a feature set |
| **Initial Vocabulary** | single bytes | single-feature sets |
| **Merging Unit** | adjacent byte pairs | feature pairs within one set or between adjacent sets |
| **Co-occurrence Weighting** | raw frequency counting | probabilistic weighting (Figure 2) |
| **Segmentation Strategy** | greedy fixed-order merging | set permutation regularization (Algorithm 4) |
| **Intermediate Structures** | N/A | intermediate nodes for cross-action merges |

when TIGER (Rajput et al., 2023) is the GR backbone). We then combine these ranking lists by averaging the scores of each predicted item. This approach applies data-level ensembling, which has been shown to enhance recommendation performance, as discussed in Section 4.4.3.

### 3.4. Discussion

**Orders in action sequences.** In recommender systems, user actions are typically represented as sets of features, such as the title and price of the associated item. These features typically have no inherent order within a single action. However, in sequential recommendation tasks, a user's historical actions are usually ordered by timestamp to capture temporal behavioral dynamics. Building on this, we model action sequences as sequences of feature sets: while the features within each action remain unordered, the temporal ordering of actions is preserved. This evolving composition of features over time captures meaningful sequential patterns.

**ActionPiece vs. BPE.** While ActionPiece follows a similar algorithmic framework as BPE, the key distinction lies in the data formats they are designed to model. BPE operates on one-dimensional byte sequences, whereas ActionPiece is tailored for tokenizing sequences of feature sets. Modeling each action as an unordered set aligns better with the inherent structure of action sequences. For clarity, we summarize the key differences in Table 2.

**Efficiency impact of SPR.** Despite introducing set permutation regularization, the training efficiency remains comparable to existing methods such as TIGER. Feature permutation is performed on the CPU and runs asynchronously alongside TPU/GPU-based model updates, resulting in no noticeable degradation in training speed. At inference time, SPR introduces additional FLOPs due to the ensemble of augmented test cases. However, the overall latency remains comparable to baseline methods, as the augmented versions can be processed in parallel across multiple computing devices (*e.g.,* TPUs or GPUs). This parallelism offsets the added computation, enabling our method to maintain efficient inference despite the use of inference-time ensembling.

*Table 3.* Statistics of the processed datasets. "Avg. $t$" denotes the average number of actions in an action sequence.

| Datasets | #Users | #Items | #Actions | Avg. $t$ |
|---|---|---|---|---|
| **Sports** | 35,598 | 18,357 | 260,739 | 8.32 |
| **Beauty** | 22,363 | 12,101 | 176,139 | 8.87 |
| **CDs** | 75,258 | 64,443 | 1,022,334 | 14.58 |

## 4. Experiments

### 4.1. Experimental Setup

**Datasets.** We use three categories from the Amazon Reviews dataset (McAuley et al., 2015) for our experiments: "Sports and Outdoors" (**Sports**), "Beauty" (**Beauty**), and "CDs and Vinyl" (**CDs**). Each user's historical reviews are considered "actions" and are sorted chronologically as action sequences, with earlier reviews appearing first. To evaluate the models, we adopt the widely used leave-last-out protocol (Kang & McAuley, 2018; Zhao et al., 2022; Rajput et al., 2023), where the last item and second-to-last item in each action sequence are used for testing and validation, respectively. The statistics of the processed datasets are shown in Table 3. More details about the datasets can be found in Appendix F.

**Compared methods.** We compare the performance of ActionPiece with the following methods: (1) ID-based sequential recommendation methods, including BERT4Rec (Sun et al., 2019), and SASRec (Kang & McAuley, 2018); (2) feature-enhanced sequential recommendation methods, such as FDSA (Zhang et al., 2019), S³-Rec (Zhou et al., 2020), and VQ-Rec (Hou et al., 2023); and (3) generative recommendation methods, including P5-CID (Hua et al., 2023), TIGER (Rajput et al., 2023), LMIndexer (Jin et al., 2024), HSTU (Zhai et al., 2024), and SPM-SID (Singh et al., 2024), each representing a different action tokenization method (Table 1). A detailed description of these baselines is provided in Appendix G.

**Evaluation settings.** Following Rajput et al. (2023), we use Recall@$K$ and NDCG@$K$ as metrics to evaluate the meth-

*Table 4.* Performance comparison of different methods on the Amazon Reviews dataset (McAuley et al., 2015). The best and second-best performance is denoted in **bold** and underlined fonts. "R@K" and "N@K" are short for "Recall@K" and "NDCG@K", respectively. "Improv." denotes the percentage improvement of our method compared to the strongest baseline method.

| Datasets | Metric | ID-based | | Feature + ID | | | Generative | | | | | | Improv. |
|---|---|---|---|---|---|---|---|---|---|---|---|---|---|
| | | BERT4Rec | SASRec | FDSA | S³-Rec | VQ-Rec | P5-CID | TIGER | LMIndexer | HSTU | SPM-SID | ActionPiece | |
| Sports | R@5 | 0.0115 | 0.0233 | 0.0182 | 0.0251 | 0.0181 | 0.0287 | 0.0264 | 0.0222 | 0.0258 | 0.0280 | **0.0316** ± 0.0005 | +12.86% |
| | N@5 | 0.0075 | 0.0154 | 0.0122 | 0.0161 | 0.0132 | 0.0179 | 0.0181 | 0.0142 | 0.0165 | 0.0180 | **0.0205** ± 0.0002 | +11.71% |
| | R@10 | 0.0191 | 0.0350 | 0.0288 | 0.0385 | 0.0251 | 0.0426 | 0.0400 | — | 0.0414 | 0.0446 | **0.0500** ± 0.0007 | +12.11% |
| | N@10 | 0.0099 | 0.0192 | 0.0156 | 0.0204 | 0.0154 | 0.0224 | 0.0225 | — | 0.0215 | 0.0234 | **0.0264** ± 0.0003 | +12.82% |
| Beauty | R@5 | 0.0203 | 0.0387 | 0.0267 | 0.0387 | 0.0434 | 0.0468 | 0.0454 | 0.0415 | 0.0469 | 0.0475 | **0.0511** ± 0.0014 | +7.58% |
| | N@5 | 0.0124 | 0.0249 | 0.0163 | 0.0244 | 0.0311 | 0.0315 | 0.0321 | 0.0262 | 0.0314 | 0.0321 | **0.0340** ± 0.0011 | +5.92% |
| | R@10 | 0.0347 | 0.0605 | 0.0407 | 0.0647 | 0.0741 | 0.0701 | 0.0648 | — | 0.0704 | 0.0714 | **0.0775** ± 0.0017 | +4.59% |
| | N@10 | 0.0170 | 0.0318 | 0.0208 | 0.0327 | 0.0372 | 0.0400 | 0.0384 | — | 0.0389 | 0.0399 | **0.0424** ± 0.0011 | +6.00% |
| CDs | R@5 | 0.0326 | 0.0351 | 0.0226 | 0.0213 | 0.0314 | 0.0505 | 0.0492 | — | 0.0417 | 0.0509 | **0.0544** ± 0.0005 | +6.88% |
| | N@5 | 0.0201 | 0.0177 | 0.0137 | 0.0130 | 0.0209 | 0.0326 | 0.0329 | — | 0.0275 | 0.0337 | **0.0359** ± 0.0004 | +6.53% |
| | R@10 | 0.0547 | 0.0619 | 0.0378 | 0.0375 | 0.0485 | 0.0785 | 0.0748 | — | 0.0638 | 0.0778 | **0.0830** ± 0.0008 | +5.73% |
| | N@10 | 0.0271 | 0.0263 | 0.0186 | 0.0182 | 0.0264 | 0.0416 | 0.0411 | — | 0.0346 | 0.0424 | **0.0451** ± 0.0005 | +6.37% |

ods, where $K \in \{5, 10\}$. Model checkpoints with the best performance on the validation set are used for evaluation on the test set. We run the experiments with five random seeds and report the average metrics.

**Implementation details.** Please refer to Appendix H for detailed implementation and hyperparameter settings.

### 4.2. Overall Performance

We compare ActionPiece with sequential recommendation and generative recommendation baselines, which use various action tokenization methods, across three public datasets. The results are shown in Table 4.

For the compared methods, we observe that those using item features generally outperform item ID-only methods. This indicates that incorporating features enhances recommendation performance. Among the methods leveraging item features ("Feature + ID" and "Generative"), generative recommendation models achieve better performance. These results further confirm that injecting semantics into item indexing and optimizing at a sub-item level enables generative models to better use semantic information and improve recommendation performance. Among all the baselines, SPM-SID achieves the best results. By incorporating the SentencePiece model (Kudo & Richardson, 2018), SPM-SID replaces popular semantic ID patterns within each item with new tokens, benefiting from a larger vocabulary.

Our proposed ActionPiece consistently outperforms all baselines across three datasets, achieving a significant improvement in NDCG@10. It surpasses the best-performing baseline method by 6.00% to 12.82%. Unlike existing methods, ActionPiece is the first context-aware action sequence tokenizer, *i.e.,* the same action can be tokenized into different tokens depending on its surrounding context. This allows ActionPiece to capture important sequence-level feature

*Table 5.* Ablation analysis of ActionPiece. The recommendation performance is measured using NDCG@10. The best performance is denoted in **bold** fonts.

| Variants | Sports | Beauty | CDs |
|---|---|---|---|
| *TIGER with varying vocabulary sizes* | | | |
| (1.1) TIGER - 192 ($4 \times 48$) | 0.0231 | 0.0362 | N/A[†] |
| (1.2) TIGER - 768 ($3 \times 2^8$) | 0.0220 | 0.0378 | 0.0331 |
| (1.3) TIGER - 1k ($4 \times 2^8$) | 0.0225 | 0.0384 | 0.0411 |
| (1.4) TIGER-49k ($6 \times 2^{13}$) | 0.0162 | 0.0317 | 0.0338 |
| (1.5) TIGER-66k ($4 \times 2^{14}$) | 0.0194 | N/A[‡] | 0.0319 |
| *Vocabulary construction* | | | |
| (2.1) *w/o* tokenization | 0.0215 | 0.0389 | 0.0346 |
| (2.2) *w/o* context-aware | 0.0258 | 0.0416 | 0.0429 |
| (2.3) *w/o* weighted counting | 0.0257 | 0.0412 | 0.0435 |
| *Set permutation regularization* | | | |
| (3.1) only for inference | 0.0192 | 0.0316 | 0.0329 |
| (3.2) only for training | 0.0244 | 0.0387 | 0.0422 |
| (3.3) TIGER + SPR | 0.0202 | 0.0330 | 0.0351 |
| ActionPiece (40k) | **0.0264** | **0.0424** | **0.0451** |

[†] not applicable because the number of conflicts among semantic ID prefixes (the first three tokens) in CDs exceeds 48.
[‡] not applicable because $2^{14}$ is larger than #items in Beauty.

patterns that enhance recommendation performance.

### 4.3. Ablation Study

We conduct ablation analyses in Table 5 to study how each proposed technique contributes to ActionPiece.

(1) To examine whether the performance gain of ActionPiece stems from the choice of vocabulary size, we conduct an ablation study by varying the vocabulary size of TIGER. We increase the number of semantic ID digits per item ($4 \rightarrow 6$) and the number of candidate semantic IDs per digit ($2^8 \rightarrow 2^{13}$ or $2^{14}$), resulting in two TIGER variants

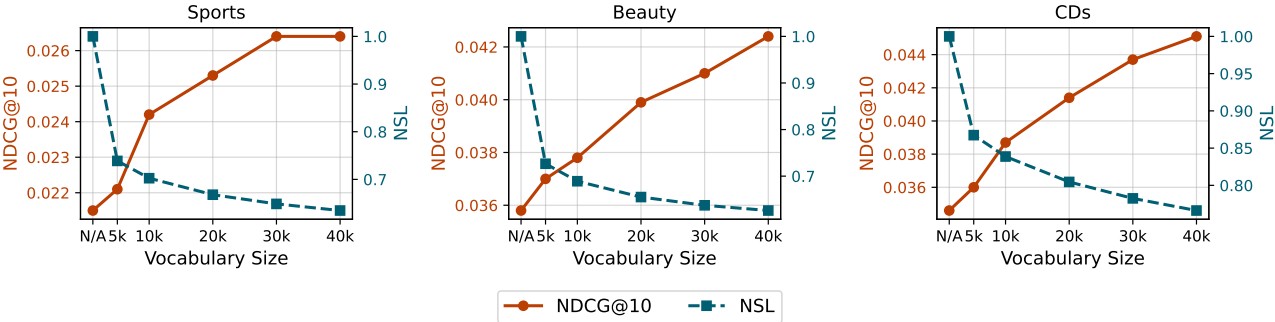

*Figure 4.* Analysis of recommendation performance (NDCG@10, ↑) and average tokenized sequence length (NSL, ↓) w.r.t. vocabulary size across three datasets. "N/A" indicates that ActionPiece is not applied, *i.e.,* action sequences are represented solely by initial tokens.

| | | |
|---|---|---|
| ///////////56.89% | | Naive |
| ///////////////87.01% | | SPR (#epochs=1) |
| //////////////////95.33% | | SPR (#epochs=5) |

Token Utilization Rate (%)

*Figure 5.* Analysis of token utilization rate (%) during model training w.r.t. segmentation strategy.

with vocabularies larger than ActionPiece. We also include two variants with reduced vocabulary sizes by decreasing the number of digits or candidates per digit. Despite the broader range of vocabulary sizes, all TIGER variants perform worse than ActionPiece, and in some cases, even worse than the original TIGER with 1,024 tokens. These results suggest that the performance improvement of ActionPiece is not simply due to scaling the vocabulary size up or down. Instead, they highlight the difficulty of effectively scaling vocabulary size in generative recommendation models, consistent with the observations from Zhang et al. (2024).

(2) To evaluate the effectiveness of the proposed vocabulary construction techniques, we introduce the following variants: *(2.1) w/o tokenization*, which skips vocabulary construction, using item features directly as tokens; *(2.2) w/o context-aware*, which only considers co-occurrences and merges tokens within each action during vocabulary construction and segmentation; and *(2.3) w/o weighted counting*, which treats all token pairs equally rather than using the weights defined in Equations (1) and (2). The results indicate that removing any of these techniques reduces performance, demonstrating the importance of these methods for building a context-aware tokenizer.

(3) To evaluate the effectiveness of SPR, we revert to naive segmentation, as described in Section 3.2.2, during model training and inference, respectively. The results show that replacing SPR with naive segmentation in either training or inference degrades performance. We also introduce an ablation variant that applies SPR to the existing GR model TIGER. The results indicate that SPR alone is insufficient to improve the GR model. When applied to action sequences tokenized by context-independent methods, SPR does not change token frequencies and merely disrupts the internal to-

ken order within RQ-VAE-based semantic IDs. In contrast, ActionPiece derives different tokens for the same item based on its neighboring context, improving token utilization and serving as an effective form of data augmentation.

### 4.4. Further Analysis

#### 4.4.1. PERFORMANCE AND EFFICIENCY W.R.T. VOCABULARY SIZE

Vocabulary size is a key hyperparameter for language tokenizers (Meta AI, 2024; Dagan et al., 2024). In this study, we investigate how adjusting vocabulary size affects the generative recommendation models. We use the normalized sequence length (NSL) (Dagan et al., 2024) to measure the length of tokenized sequences, where a smaller NSL indicates fewer tokens per tokenized sequence. We experiment with vocabulary sizes in {N/A, 5k, 10k, 20k, 30k, 40k}, where "N/A" represents the direct use of item features as tokens. As shown in Figure 4, increasing the vocabulary size improves recommendation performance and reduces the tokenized sequence length. Conversely, reducing the vocabulary size lowers the number of model parameters, improving memory efficiency. This analysis demonstrates that adjusting vocabulary size enables a trade-off between model performance, sequence length, and memory efficiency.

#### 4.4.2. TOKEN UTILIZATION RATE W.R.T. SEGMENTATION STRATEGY

As described in Section 3.3.1, applying SPR augments the training corpus by producing multiple token sequences that share the same semantics. In Table 5, we observe that incorporating SPR significantly improves recommendation performance. One possible reason is that SPR increases token utilization rates. To validate this assumption, we segment the action sequences in each training epoch using two strategies: naive segmentation and SPR. As shown in Figure 5, naive segmentation uses only 56.89% of tokens for model training, limiting the model's ability to generalize to unseen action sequences. In contrast, SPR achieves a

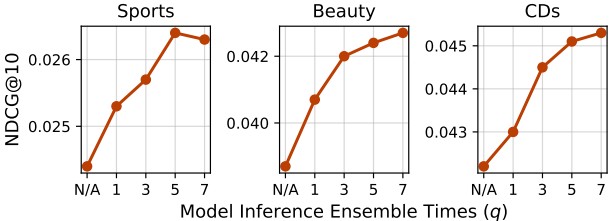

*Figure 6.* Analysis of performance (NDCG@10, ↑) w.r.t. the number of ensembled segments $q$ during model inference.

token utilization rate of $87.01\%$ after the first training epoch, with further increases as training progresses. These results demonstrate that the proposed SPR segmentation strategy improves the utilization of ActionPiece tokens, enabling better generalization and enhanced performance.

### 4.4.3. PERFORMANCE W.R.T. INFERENCE-TIME ENSEMBLES

As described in Section 3.3.2, ActionPiece supports inference-time ensembling by using SPR segmentation. We vary the number of ensembled segments, $q$, in {N/A, 1, 3, 5, 7}, where "N/A" indicates using naive segmentation during model inference. As shown in Figure 6, ensembling more tokenized sequences improves ActionPiece's recommendation performance. However, the performance gains slow down as $q$ increases to 5 and 7. Since a higher $q$ also increases the computational cost of inference, this creates a trade-off between performance and computational budget in practice.

### 4.5. Case Study

To understand how GR models benefit from the unordered feature setting and context-aware action sequence tokenization, we present an illustrative example in Figure 1.

Each item in the action sequence is represented as a feature set, with each item consisting of five features. The features within an item do not require a specific order. The first step of tokenization leverages the unordered nature of the feature set and applies set permutation regularization (Section 3.2.2). This process arranges each feature set into a specific permutation and iteratively groups features based on the constructed vocabulary (Section 3.2.1). This results in different segments that convey the same semantics. Each segment is represented as a sequence of sets, where each set corresponds to a token in the vocabulary.

By examining the segments and their corresponding token sequences, we identify four types of tokens, as annotated in Figure 1: (1) a subset of features from a single item (token 14844 corresponds to features 747 and 923 of the T-shirt); (2) a set containing a single feature (feature 76 of the socks); (3) all features of a single item (token 7995 corresponds to all features of the shorts); and (4) features from multiple items (*e.g.,* token 8316 includes feature 923 from the T-

shirt and feature 679 from the socks, while token 19895 includes feature 1100 from the socks as well as features 560 and 943 from the shorts). Notably, the fourth type of token demonstrates that the features of one action can be segmented and grouped with features from adjacent actions. This results in different tokens for the same action depending on the surrounding context, showcasing the context-aware tokenization process of ActionPiece.

## 5. Conclusion

In this paper, we introduce ActionPiece, the first context-aware action sequence tokenizer for generative recommendation. By considering the surrounding context, the same action can be tokenized into different tokens in different sequences. We formulate generative recommendation as a task on sequences of feature sets and merge important feature patterns into tokens. During vocabulary construction, we propose assigning weights to token pairs based on their structures, such as those within a single set or across adjacent sets. To enable efficient vocabulary construction, we use double-ended linked lists to maintain the corpus and introduce intermediate nodes to store tokens that combine features across adjacent sets. Additionally, we propose set permutation regularization, which segments a single action sequence into multiple token sequences with the same semantics. These segments serve as natural augmentations for training and as ensemble instances for inference.

In the future, we plan to align user actions with other modalities by constructing instructions that combine ActionPiece tokens and other types of tokens. We also aim to extend the proposed tokenizer to other tasks that can be framed as set sequence modeling problems, including audio modeling, sequential decision-making, and time series forecasting.

## Acknowledgements

This work was conducted while Yupeng Hou was a student researcher at Google DeepMind. We thank Yichen Zhou for helpful discussions and Nikhil Mehta for valuable suggestions on the draft.

## Impact Statement

This paper introduces ActionPiece, a context-aware action sequence tokenizer to enhance generative recommendation. This work aims to advance personalized recommender systems, enabling more accurate understanding of user preferences. We argue that this work is not directly correlated to certain society or ethical concerns. The impact of this work is primarily tied to the broader implications of recommender systems in various domains, such as e-commerce, entertainment, and social platforms.

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

*Table 6.* Notations and explanations.

| Notation | Explaination |
| --- | --- |
| $i, i_1, i_j$ | Item, item identifier, item ID |
| $t$ | The number of actions in the input action sequence; the timestamp when the model makes a prediction |
| $i_{t+1}$ | The ground-truth next item |
| $\hat{i}_{t+1}$ | The predicted next item |
| $S = \{i_1, i_2, \dots, i_t\}$ | The action sequence where each action is represented with the interacted item ID |
| $\mathcal{A}, \mathcal{A}_1, \mathcal{A}_j$ | A set of item features or tokens |
| $m = |\mathcal{A}_j|$ | The number of features associated with each item |
| $f_{j,k}$ | The $k$-th feature of item $i_j$ |
| $\mathcal{F}_k$ | The collection of all possible choices for the $k$-th feature |
| $S' = \{\mathcal{A}_1, \mathcal{A}_2, \dots, \mathcal{A}_t\}$ | The action sequence where each action is represented with a set of item features |
| $c, c_1, c_j$ | Input & generated tokens |
| $l$ | The number of tokens in the token sequence |
| $C = \{c_1, c_2, \dots, c_l\}$ | The token sequence tokenized from the input action sequence $S'$ |
| $\{c_{l+1}, \dots, c_q\}$ | The tokens generated by the GR model |
| $\mathcal{V}$ | The vocabulary of ActionPiece tokenizer |
| $\mathcal{R}$ | The merge rules of ActionPiece tokenizer |
| $\{(c_u, c_v) \rightarrow c_{\text{new}}\}$ | One merge rule indicating two adjacent tokens $c_u$ and $c_v$ can be replaced by a token $c_{\text{new}}$ |
| $Q = |\mathcal{V}|$ | The size of ActionPiece vocabulary |
| $P(c, c')$ | The probability that tokens $c$ and $c'$ are adjacent when flattening a sequence of sets into a token sequence |
| $N$ | The number of action sequences in the training corpus |
| $L$ | The average length of action sequences in the training corpus |
| $H$ | Maximal heap size, $O(NLm)$ |
| $q$ | The number of segmentations produced using set permutation regularization during inference |

# Appendices

## A. Notations

We summarize the notations used in this paper in Table 6.

## B. Algorithmic Details

In this section, we provide detailed algorithms for vocabulary construction and segmentation.

### B.1. Vocabulary Construction Algorithm

The overall procedure for vocabulary construction is illustrated in Algorithm 1. As described in Section 3.2.1, this process involves iterative **Count** (Algorithm 2) and **Update** (Algorithm 3) operations.

### B.2. Segmentation with Set Permutation Regularization Algorithm

The detailed algorithm for segmenting action sequences into token sequences using set permutation regularization (SPR) is shown in Algorithm 4. In practice, we often run Algorithm 4 multiple times to augment the training corpus or ensemble recommendation outputs, as described in Sections 3.3.1 and 3.3.2.

## C. Efficient Vocabulary Construction Implementation

To efficiently construct the ActionPiece vocabulary, we propose using data structures such as heaps with a lazy update trick, linked lists, and inverted indices to speed up each iteration of the construction process. The key idea is to avoid recalculating token co-occurrences in every iteration and instead update the data structures. The pseudocode is shown in Figure 7.

---

**Algorithm 2** ActionPiece Vocabulary Construction – Count (Figure 2)

---

**input** Action sequence corpus $\mathcal{S}'$, current vocabulary $\mathcal{V}$
**output** Accumulated weighted token co-occurrences $\mathrm{count}(\cdot, \cdot)$

1: **for** $i \leftarrow 0$ **to** $|\mathcal{V}|, j \leftarrow 0$ **to** $|\mathcal{V}|$ **do**
2:    $\mathrm{count}(c_i, c_j) \leftarrow 0$
3: **end for**
4: **for all** sequence $S' \in \mathcal{S}'$ **do**
5:    $t \leftarrow \mathrm{length}(S')$ # Number of action nodes in sequence
6:    **for** $k \leftarrow 0$ **to** $t - 1$ **do**
7:       $\mathcal{A}_k \leftarrow S'[k]$ # Current action node
8:       # Process all unordered token pairs within $\mathcal{A}_k$
9:       **for all** $c_i, c_j \in \mathcal{A}_k, i \neq j$ **do**
10:          $\mathrm{count}(c_i, c_j) \leftarrow \mathrm{count}(c_i, c_j) + 2/|\mathcal{A}_k|$ # Weight of tokens within a single set (Equation (1))
11:          $\mathrm{count}(c_j, c_i) \leftarrow \mathrm{count}(c_j, c_i) + 2/|\mathcal{A}_k|$ # Symmetric update
12:       **end for**
13:       # Process all ordered token pairs between $A_k$ and $A_{k+1}$
14:       **if** $k < t - 1$ **then**
15:          $\mathcal{A}_{k+1} \leftarrow S'[k+1]$
16:          **for all** $c_i \in \mathcal{A}_k, c_j \in \mathcal{A}_{k+1}$ **do**
17:             $\mathrm{count}(c_i, c_j) \leftarrow \mathrm{count}(c_i, c_j) + 1/(|\mathcal{A}_k| \times |\mathcal{A}_{k+1}|)$ # Weight of tokens from two adjacent sets (Equation (2))
18:          **end for**
19:       **end if**
20:    **end for**
21: **end for**
**return** $\mathrm{count}(\cdot, \cdot)$

---

## C.1. Data Structures

The data structures used in the proposed algorithm are carefully designed to optimize the efficiency of vocabulary construction. Here is a detailed discussion of their roles and implementations:

- **Linked list:** Each action sequence in the training corpus is stored as a linked list. This allows efficient local updates during token merging. When a token pair $(c_u, c_v)$ is replaced by a new token $c_{new}$, only the affected nodes and their neighbors in the linked list need to be modified (as shown in Algorithm 3 and Figure 3).

- **Heap with lazy update trick:** A max-heap prioritizes token pairs by their co-occurrences. Instead of recalculating the heap entirely in each iteration, a "lazy update" strategy is employed: outdated entries (with mismatched co-occurrence counts) are retained but skipped during extraction. In the pseudocode, the loop checks if the top element is outdated via `is_outdated`. Invalid entries are discarded, and only valid ones are processed. Updated co-occurrences are pushed as new entries (with negative counts for max-heap emulation).

- **Inverted indices:** The `pair2head` dictionary maps token pairs to the sequences containing them. When a pair $(c_u, c_v)$ is merged, the algorithm directly retrieves affected sequence IDs via `pair2head[(c_u, c_v)]`, avoiding a full corpus scan. After merging, the inverted indices are incrementally updated: new token pairs (*e.g.*, $(c_{prev}, c_{new})$ and $(c_{new}, c_{next})$) are added to `pair2head`, while obsolete pairs are removed. This enables targeted updates and ensures subsequent iterations efficiently access relevant sequences.

These structures collectively reduce time complexity by focusing computation on dynamically changing parts of the corpus and avoiding redundant global operations. The linked list enables localized edits, the heap minimizes priority recalculation, and the inverted indices eliminate brute-force searches, making the algorithm scalable to large corpora.

## C.2. Time Complexity

The time complexity of the efficient vocabulary construction algorithm can be analyzed through two main components: **initialization** and **iterative merging**.

**Algorithm 3** ActionPiece Vocabulary Construction – Update (Figure 3)

---

**input** Action sequence corpus $\mathcal{S}'$ before updating, current merge rule $\{(c_u, c_v) \rightarrow c_{\text{new}}\}$
**output** Updated action sequence corpus $\mathcal{S}'$

1: **for all** sequence $S' \in \mathcal{S}'$ **do**
2:      $t \leftarrow \text{length}(S')$
3:      **for** $k \leftarrow 0$ **to** $t - 1$ **do**
4:          $\mathcal{A}_k \leftarrow S'[k]$
5:          # Merge tokens in one action node
6:          **if** $c_u \in \mathcal{A}_k$ **and** $c_v \in \mathcal{A}_k$ **then**
7:             Replace $c_u$ and $c_v$ in $\mathcal{A}_k$ with $c_{\text{new}}$
8:          **end if**
9:          # Merge tokens from two adjacent nodes
10:         **if** $k < t - 1$ **then**
11:            $\mathcal{A}_{k+1} \leftarrow S'[k + 1]$
12:            **if** $c_u \in \mathcal{A}_k$ **and** $c_v \in \mathcal{A}_{k+1}$ **then**
13:               **if** $\mathcal{A}_k, \mathcal{A}_{k+1}$ are both action nodes **then**
14:                  Create intermediate node $M$ between $\mathcal{A}_k$ and $\mathcal{A}_{k+1}$
15:                  $M \leftarrow \{c_{\text{new}}\}$ # Linked list: $\mathcal{A}_k \rightarrow M \rightarrow \mathcal{A}_{k+1}$
16:                  $\mathcal{A}_k \leftarrow \mathcal{A}_k \setminus c_u$
17:                  $\mathcal{A}_{k+1} \leftarrow \mathcal{A}_{k+1} \setminus c_v$
18:               **else if** $\mathcal{A}_k$ is intermediate node **then**
19:                  $\mathcal{A}_k \leftarrow \{c_{\text{new}}\}$
20:                  $\mathcal{A}_{k+1} \leftarrow \mathcal{A}_{k+1} \setminus c_v$
21:               **else if** $\mathcal{A}_{k+1}$ is intermediate node **then**
22:                  $\mathcal{A}_k \leftarrow \mathcal{A}_k \setminus c_u$
23:                  $\mathcal{A}_{k+1} \leftarrow \{c_{\text{new}}\}$
24:               **end if**
25:            **end if**
26:         **end if**
27:      **end for**
28: **end for**
**return** $\mathcal{S}'$

---

- **Initialization phase** involves building the initial max-heap to track co-occurrence frequencies. Given $N$ input sequences (each with an average length of $L$), we count co-occurrences for all $O(m^2)$ token pairs within each set of size $m$. This requires $O(NLm^2)$ time.

- **Iterative merging phase** dynamically processes the involved sequences. The total number of such sequences across all iterations is approximately

$$O\left(\frac{N}{|\mathcal{V}_0|}\right) + O\left(\frac{N}{|\mathcal{V}_0| + 1}\right) + \cdots + O\left(\frac{N}{Q}\right) \simeq O(\log QN).$$

For each sequence, updating the linked list requires $O(Lm)$ time, counting co-occurrences takes $O(Lm^2)$ time, and inserting co-occurrences into the max-heap requires at most $O(Lm^2 \log H)$ time. Here, $H$ represents the heap size, which is at most $O(NLm)$. Thus, the overall time complexity for iterative merging is

$$O(\log QN(Lm + Lm^2 + Lm^2 \log H)) = O(\log Q \log H \cdot NLm^2).$$

Therefore, the overall time complexity of our proposed vocabulary construction algorithm is $O(\log Q \log H \cdot NLm^2)$, where the iterative merging phase dominates. This complexity is significantly better than the naive vocabulary construction complexity of $O(QNLm^2)$.

---

**Algorithm 4** Segmentation via Set Permutation Regularization (SPR) (Section 3.2.2)

---

**input** Action sequence $S$, merge rules $\mathcal{R}$
**output** Segmented token sequences $C$

1:   $C \leftarrow [\,]$ # Initialize permuted initial token sequence
2:   **for all** token set $\mathcal{A}_i \in S$ **do**
3:      Generate random permutation of $\mathcal{A}_i$ as $[c_1, c_2, \ldots, c_{|\mathcal{A}_i|}]$
4:      Extend $C$ with $[c_1, c_2, \ldots, c_{|\mathcal{A}_i|}]$ # Concatenate permutations
5:   **end for**
6:
7:   # Apply BPE (Sennrich et al., 2016) segmentation on permuted sequence
8:   **repeat**
9:      $\mathcal{R}' \leftarrow \emptyset$ # Candidate merge rules
10:     **for** $i \leftarrow 0$ **to** $|C| - 1$ **do**
11:        **if** $\{(c_i, c_{i+1}) \rightarrow c'\} \in \mathcal{R}$ **then**
12:           $\mathcal{R}' \leftarrow \mathcal{R}' \cup \{(c_i, c_{i+1}) \rightarrow c'\}$
13:        **end if**
14:     **end for**
15:     Select $\{(c_k, c_{k+1}) \rightarrow c'\} \in \mathcal{R}'$ with the smallest index among all merge rules $\mathcal{R}$
16:     $C \leftarrow [c_1, \ldots, c_{k-1}, c', c_{k+2}, \ldots]$ # Replace $(c_k, c_{k+1})$ with a new token $c'$
17: **until** $\mathcal{R}'$ is $\emptyset$
**return** $C$

---

# D. Additional Related Work

**Aligning LLMs with user actions.** A key motivation for developing action tokenization methods is to provide an efficient and effective way of aligning pretrained generative models (*e.g.,* LLMs (OpenAI, 2022; Anil et al., 2023; Touvron et al., 2023; Zhao et al., 2023)) with user action data (Geng et al., 2022; Zheng et al., 2024; Hou et al., 2025). A typical pipeline involves tokenizing action sequences into action tokens and then performing instruction tuning on inputs that combine both language and action tokens. The choice of tokenization strategy plays a critical role. We identify three main paradigms:

- *One token per item*: Each action is represented by a single token, resulting in a dense vector that is typically derived from a pretrained semantic encoder or a learnable embedding table (Hou et al., 2022; Liao et al., 2024; Zhu et al., 2024b; Zhang et al., 2025b; Kim et al., 2024). While this approach is efficient in terms of sequence length, it suffers from memory and scalability issues, particularly since the number of unique items often exceeds the typical vocabulary size of LLMs. Aligning LLMs with such large vocabularies introduces both engineering and optimization challenges.

- *Text-based tokenization*: Each action is expressed as a textual string, which naturally aligns with the LLMs' native modality (Geng et al., 2022; Zhang et al., 2025a; Bao et al., 2023). However, this leads to substantially longer token sequences, resulting in inefficiencies during tokenization and increased inference latency.

- *Discrete tokens*: Actions are tokenized into a small number of discrete tokens drawn from a compact, shared vocabulary (Zheng et al., 2024; Tan et al., 2024; Jin et al., 2024). This approach balances sequence length and memory efficiency, making it a practical choice for building LLM-based recommender systems.

# E. Additional Discussions

## E.1. Benefits of Set Permutation Regularization

SPR benefits the model from multiple perspectives:

- *Token utilization perspective.* SPR effectively prevents the features of a single action from being consistently merged into the most compressed (high-level) tokens. Instead, it allows the action to be tokenized into both high-level and low-level tokens, depending on the permutation and token merge rules. This increases the number of tokens actively involved during both training and inference. As shown in Figure 5 and discussed in Section 4.4.2, SPR significantly

```python
def vocab_construction_iteration(max_heap, vocab, rules, pair2head):
    """Performs one iteration of efficient vocabulary construction.

    Args:
        max_heap (PriorityQueue): Max-heap storing (co_occurrence, (c_u, c_v)) pairs.
        vocab (List[Tuple]): Current vocabulary with merge rules.
        rules (Dict): Merge rule {(c_u, c_v): c_new} mapping.
        pair2head (Dict): Inverted indices that store mappings from the token pair to
            all sequences that contain this token pair.
    """
    # Get most frequent valid pair (c_u, c_v) from max-heap
    # Efficient version of "Count" in Algorithm 1
    # Avoid recalculating co-occurrences for each iteration, by maintaining them in a
        max-heap with the lazy update trick
    while not max_heap.empty():
        co_occurrence, (c_u, c_v) = max_heap.get()
        if not is_outdated((c_u, c_v), co_occurrence):  # Outdated values are lazily
            removed.
            break

    # Create new token and update vocabulary
    c_new = len(vocab)
    vocab.append((c_u, c_v))
    rules[(c_u, c_v)] = c_new

    # Update sequences containing (c_u, c_v)
    seq_ids = pair2head[(c_u, c_v)].copy()  # IDs of affected sequences
    delta_counts = defaultdict(int)

    for sid in seq_ids:
        # Merge all (c_u, c_v) pairs in sequence
        new_seq = merge_sequence(seqs[sid], c_u, c_v, c_new)  # Replace pairs
        seqs[sid] = new_seq

        # Calculate pair co-occurrence changes
        new_freqs = count_pairs(new_seq)  # Get token co-occurrences of the updated
            sequence
        delta = diff_counts(new_freqs, old_freqs[sid])  # Compute co-occurrence
            differences
        update_index(pair2head, delta, sid)  # Update inverted index
        old_counts[sid] = new_freqs

        # Accumulate global co-occurrence changes
        for p, cnt in delta.items():
            delta_counts[p] += cnt

    # Lazy update max-heap with new co-occurrences
    for (c_u, c_v), delta in delta_counts.items():
        if abs(delta) < eps: continue  # eps: minimum update threshold
        all_pair_freqs[(c_u, c_v)] += delta  # Global co-occurrences
        max_heap.put( (-all_pair_freqs[(c_u, c_v)], (c_u, c_v)) )
```

*Figure 7.* Pseudocode for a single iteration of the efficient vocabulary construction algorithm, illustrating how a max-heap with lazy updates is used to track and merge frequent token pairs.

improves token utilization - from 56.89% to 95.33% by the 5th epoch - indicating that a greater proportion of tokens are trained after applying SPR.

- *Data augmentation perspective.* From the perspective of data augmentation, SPR enriches the token sequences available for model training. Without SPR, each action sequence can only be tokenized into a single, fixed token sequence. In contrast, SPR allows each action sequence to be tokenized in multiple ways (as shown in Figure 1). While these augmented sequences preserve the same semantic information, they expose the model to richer token patterns. Training on these diverse token sequences helps the model generalize better, as evidenced by the performance of variant (3.1)

in Table 5.

- *Ensemble perspective.* SPR also enables inference-time augmentation. A given input action sequence can be augmented into multiple token sequences during inference. Each sequence may yield a different ranking of the next possible items. By ensembling these recommendation results, overall performance can be enhanced, as demonstrated by variant (3.2) in Table 5 and further illustrated in Figure 6.

### E.2. Connections Between ActionPiece and Feature Crossing

A core component of ranking models is automatically capturing feature-crossing patterns that are helpful for recommendation performance (Shan et al., 2016; Wang et al., 2017; 2021; Bian et al., 2022), which is conceptually related to ActionPiece. The key difference lies in the level at which interactions are modeled. Prior works typically perform automatic feature crossing at the model level, learning interactions implicitly through the network architecture. In contrast, ActionPiece merges features at the data level.

Although we do not directly compare against these specific feature-crossing models, we include several baselines that follow a similar design philosophy. For example, HSTU in Table 4 and variant (2.1) in Table 5 use the same underlying item features as ActionPiece but input them in a flattened form, without merging. In these setups, feature interactions are expected to be learned by the autoregressive model via self-attention and feed-forward layers, *i.e.,* at the model level.

Our results demonstrate that ActionPiece, which performs data-level feature merging, consistently outperforms these methods in both recommendation quality (Tables 4 and 5) and efficiency (Figure 4). This is particularly evident in normalized sequence length (NSL): both HSTU and variant (2.1) yield NSL values of 1, reflecting longer sequences compared to the more compact sequences tokenized by ActionPiece.

## F. Datasets

**Categories.** Among all the datasets, "Sports" and "Beauty" are two widely used benchmarks for evaluating generative recommendation models (Rajput et al., 2023; Jin et al., 2024; Hua et al., 2023). We conduct experiments on these benchmarks to ensure fair comparisons with existing results. Additionally, we introduce "CDs", which contains about $4\times$ more interactions than "Sports", making it a larger dataset for evaluating the scalability of GR models. For "CDs", since there are no publicly available results from generative recommendation methods to date, we closely followed the experimental settings used in public benchmarks like "Sports" and "Beauty" to ensure fair comparisons.

**Sequence truncation length.** Following Rajput et al. (2023), we filter out users with fewer than 5 reviews and truncate action sequences to a maximum length of 20 for "Generative" methods, including ActionPiece. For "ID-based" and "Feature + ID" baselines, we set the maximum length to 50, as suggested in their original papers.

**Item text features.** Following Rajput et al. (2023); Zheng et al. (2024); Hou et al. (2024a); Sheng et al. (2025), the first step for feature engineering is to combine multiple raw text features into a single sentence for each item. Then, we use a pretrained sentence embedding model to encode this sentence into a vector representation. In all our implementations, we concatenate *title*, *price*, *brand*, *feature*, *categories*, and *description*, and use `sentence-t5-base` (Ni et al., 2022) as the sentence embedding model.

- The encoded sentence embeddings of 768 dimension are directly used as textual item representations for UniSRec.

- We quantize the sentence embeddings using residual quantization (RQ) (Rajput et al., 2023; Zeghidour et al., 2021; Zheng et al., 2025) into three codes, each with 256 candidates. To prevent conflicts, we add an extra identification code. These four codes together serve as the RQ-based semantic IDs for TIGER and SPM-SID.

- For other baselines that require item features, such as FDSA, S³-Rec, VQ-Rec, HSTU, and our method, we follow Hou et al. (2023) and quantize the sentence embeddings using optimized product quantization (OPQ) (Ge et al., 2013). Except for VQ-Rec, where the sentence embeddings are quantized into 32 codes as suggested in the original paper, we quantize the sentence embeddings into 4 codes for all other methods to ensure a fair comparison. The codebook size is 256 for each digit of code. For generative methods HSTU and ActionPiece, we also include an additional identification code to prevent conflicts. Note that, unlike RQ-based semantic IDs, features produced by product/vector quantization do not require a specific order.

# G. Baselines

We compare ActionPiece with the following representative baselines:

## G.1. ID-Based Sequential Recommendation Methods

- **SASRec** (Kang & McAuley, 2018) represents each item using its unique item ID. It encodes item ID sequences with a self-attentive Transformer decoder. The model is trained by optimizing a binary cross-entropy objective.

- **BERT4Rec** (Sun et al., 2019) also represents each item using its unique item ID. Unlike SASRec, it encodes sequences of item IDs with a bidirectional Transformer encoder. The model is trained using a masked prediction objective.

## G.2. Feature-Enhanced Sequential Recommendation Methods

- **FDSA** (Zhang et al., 2019) integrates item feature embeddings with vanilla attention layers to obtain feature representations. It then processes item ID sequences and feature sequences separately through self-attention blocks.

- **S$^3$-Rec** (Zhou et al., 2020) first employs self-supervised pre-training to capture the correlations between item features and item IDs. Then the checkpoints are loaded and fine-tuned for next-item prediction, using only item IDs.

- **VQ-Rec** (Hou et al., 2023) encodes text features into dense vectors using pre-trained language models. It then applies product quantization to convert these dense vectors into semantic IDs. The semantic ID embeddings are pooled together to represent each item. Since the experiments are not performed in a transfer learning setting, we omit the two-stage training strategy outlined in the original paper. Instead, we reuse the model architecture and train it from scratch using an in-batch contrastive loss with a batch size of 256.

## G.3. Generative Recommendation Methods

Each generative recommendation baseline corresponds to an action tokenization method described in Table 1.

- **P5-CID** (Hua et al., 2023) is an extension of P5 (Geng et al., 2022), which formulates recommendation tasks in a text-to-text format. Building on P5, the authors explored several tokenization methods to index items for better recommendations. In this study, we use P5-CID as a representative hierarchical clustering-based action tokenization method. It organizes the eigenvectors of the Laplacian matrix of user-item interactions into a hierarchy and assigns cluster IDs at each level as item indices. When implementing this baseline method, we adopt the same model backbone as ActionPiece (encoder-decoder Transformers trained from scratch) and use the indices produced by P5-CID.

- **TIGER** (Rajput et al., 2023) encodes text features similarly to VQ-Rec but quantizes them into semantic IDs using RQ-VAE. The model is then trained to autoregressively predict the next semantic ID and employs beam search for inference. We use a beam size of 50 in beam search to generate the top-$K$ recommendations.

- **LMIndexer** (Jin et al., 2024) takes text as input and predicts semantic IDs. The text description of each item is first tokenized using a text tokenizer. The resulting text tokens are then concatenated to form input action sequences. The model is trained with self-supervised objectives to learn the semantic IDs of target items. The reported results in Table 4 are taken from the original paper. We do not report the results of LMIndexer on the large dataset "CDs" because it does not converge under similar computing budget as the other methods.

- **HSTU** (Zhai et al., 2024) discretizes raw item features into tokens, treating them as input tokens for generative recommendation. The authors also propose a lightweight Transformer layer that improves both performance and efficiency. For action tokenization, we use the same item features as our method and arrange them in a specific order to form the tokenized tokens of each item.

- **SPM-SID** (Singh et al., 2024) first tokenizes each item into semantic IDs. It then uses the SentencePiece model (SPM) (Kudo & Richardson, 2018) to merge important semantic ID patterns within each item into new tokens in the vocabulary. While the original paper introduces this method for ranking models, we adapt it for the generative recommendation task. Specifically, we concatenate the SPM tokens as inputs, feed them into the T5 model, and autoregressively generate SPM tokens as recommendations.

*Table 7.* Hyperparameter settings of ActionPiece for each dataset.

| Hyperparameter | Sports | Beauty | CDs |
|---|---|---|---|
| learning_rate | 0.005 | 0.001 | 0.001 |
| warmup_steps | 10,000 | 10,000 | 10,000 |
| dropout_rate | 0.1 | 0.1 | 0.1 |
| weight_decay | 0.15 | 0.15 | 0.07 |
| vocabulary_size | 40,000 | 40,000 | 40,000 |
| n_inference_segments | 5 | 5 | 5 |
| beam_size | 50 | 50 | 50 |
| num_layers | 4 | 4 | 4 |
| d_model | 128 | 128 | 256 |
| d_ff | 1,024 | 1,024 | 2,048 |
| num_heads | 6 | 6 | 6 |
| d_kv | 64 | 64 | 64 |
| optimizer | adamw | adamw | adamw |
| lr_scheduler | cosine | cosine | cosine |
| train_batch_size | 256 | 256 | 256 |
| max_epochs | 200 | 200 | 200 |
| early_stop_patience | 20 | 20 | 20 |

## H. Implementation Details

**Baselines.** The results of BERT4Rec, SASRec, FDSA, S³-Rec, TIGER, and LMIndexer on the "Sports" and "Beauty" benchmarks are taken directly from existing papers (Zhou et al., 2020; Rajput et al., 2023; Jin et al., 2024). For other results, we carefully implement the baselines and tune hyperparameters according to the suggestions in their original papers. We implement BERT4Rec, SASRec, FDSA, and S³-Rec using the open-source recommendation library RecBole (Zhao et al., 2021). For other methods, we implement them ourselves with HuggingFace Transformers (Wolf et al., 2020) and PyTorch (Paszke et al., 2019). We use FAISS (Douze et al., 2024) to quantize sentence representations.

**ActionPiece.** We use an encoder-decoder Transformer architecture similar to T5 (Raffel et al., 2020). We use four layers for both the encoder and decoder. The multi-head attention module has six heads, each with a dimension of $64$. For the public benchmarks "Sports" and "Beauty", we follow Rajput et al. (2023) and set the token embedding dimension to $128$ and the intermediate feed-forward layer dimension to $1024$. This results in a total of 4.46M non-embedding parameters. For the larger "CDs" dataset, we use a token embedding dimension of $256$ and an intermediate feed-forward layer dimension of $2048$, leading to 13.11M non-embedding parameters. For model inference, we use beam search with a beam size of $50$. Note that the baselines P5-CID, TIGER, and SPM-SID use the same model architecture, differing only in their action tokenization methods. For ActionPiece-specific hyperparameters, we set the number of segmentations produced using set permutation regularization during inference to $q = 5$. We tune the vocabulary size in $\{5k, 10k, 20k, 30k, 40k\}$.

**Training.** We train the GR models from scratch for up to $200$ epochs, using early stopping if the model does not achieve a better NDCG@10 on the validation set for $20$ consecutive epochs. The training batch size is set to $256$. The learning rate is selected from $\{1 \times 10^{-3}, 3 \times 10^{-3}, 5 \times 10^{-3}\}$ with a warmup step of $10,000$. We use a dropout rate of $0.1$ and tune the weight decay from $\{0.07, 0.1, 0.15, 0.2\}$. For all methods implemented by us, we conduct five repeated experiments using random seeds $\{2024, 2025, 2026, 2027, 2028\}$. The model checkpoints with the best average NDCG@10 on the validation set are selected for evaluation on the test set, and we report these results. Each model is trained on a single $40$G NVIDIA A100 GPU.

**Inference.** The inference process of ActionPiece follows the same procedure as TIGER. The decoder autoregressively generates token sequences representing the target items. During training, we use the original item features as labels, without any augmentation or token merging. At inference time, for each augmented version of a test case, we apply beam search to generate top-ranked token sequences. The most probable sequences (*i.e.,* prefixes) are retained in the beam (with beam size specified in Table 7), and the model continues generating tokens one at a time until the target sequence length is reached.

