# OpenReview forum: "ActionPiece: Contextually Tokenizing Action Sequences for Generative Recommendation"
_ICML.cc/2025/Conference — ICML 2025 spotlightposter_

### Official Review · Reviewer_DB8e · 2025-03-13

**Overall Recommendation:** 4

**Summary:**

The paper addresses a common limitation in existing generative models, where actions are tokenized independently. To resolve this issue, the paper introduces ActionPiece, a novel method that explicitly incorporates contextual information when tokenizing action sequences. Experimental results on public datasets show that ActionPiece consistently outperforms existing action tokenization methods.

**Claims And Evidence:**

Yes, the claims made in the submission are supported by clear and convincing evidence.

**Essential References Not Discussed:**

LLMs-based sequence recommendation methods.

**Experimental Designs Or Analyses:**

The experimental design of the paper is reasonable and well-structured. The authors conduct comprehensive ablation studies that effectively validate the importance of constructing a context-aware tokenizer.

**Methods And Evaluation Criteria:**

Yes, the proposed methods and evaluation criteria are well-suited for the problem at hand.

**Other Comments Or Suggestions:**

1. Exploring LLMs-based sequence recommendation methods could provide a broader perspective on the potential applications and improvements of the proposed technique.

2. Evaluating the training on cross-domain datasets could help demonstrate its robustness and generalization across different use cases.

**Other Strengths And Weaknesses:**

Strengths:

1. The motivation behind the paper is well-founded, and the proposed method is novel. It introduces the innovative idea of considering encoding contextual information, which allows for the inclusion of more features to be learned together, enhancing the effectiveness of the tokenization process.
2. The paper is logically rigorous in its writing, with a thorough experimental process that supports the claims made in the study.

Weaknesses:
1. The paper does not discuss other sequence recommendation methods that use large language models (LLMs) as backbones.

2. The paper does not attempt cross-dataset pretraining to validate the generalization ability of the model.

**Questions For Authors:**

1. When computing token co-occurrence statistics, why are token pairs between adjacent actions assigned lower weights?
2. Would the recommendation performance improve as the number of parameters in the backbone increases? Could you provide experimental validation for this?

**Relation To Broader Scientific Literature:**

This paper introduces and extends previous encoding methods used in recommendation tasks, enhancing the richness of the encoded content. The tokenization method proposed in the paper is, to some extent, inspired by research in the field of natural language processing (NLP).

**Theoretical Claims:**

The paper does not provide proofs for the theoretical claims it presents.

---

> ### Author Rebuttal · Authors · 2025-03-31
>
> We thank the reviewer for their thoughtful suggestions! Below, we address the questions listed under "Questions for Authors", followed by further discussion on related topics.
>
> **Q1: Why are token pairs between adjacent actions assigned lower weights?**
>
> **A1:** First, we'd like to clarify that token pairs between adjacent actions are not always assigned lower weights compared to token pairs within an action.
>
> For example, consider an action A with 8 features and an adjacent action B with 3 features. Then:
> * The weight for token pairs between actions A and B is:
> $\frac{1}{8 \times 3} = \frac{1}{24}$
> * The weight for token pairs within action A is:
> $\frac{1}{\binom{8}{2}} = \frac{1}{28} < \frac{1}{24}$
>
> This illustrates that in some cases, cross-action token pairs actually receive higher weights than within-action pairs.
>
> More broadly, the weighting scheme is designed to reflect the expected probability that two tokens co-occur as neighbors in the flattened sequence, where tokens in each action (set) are randomly permuted. As a result, the final weight depends on the size of the involved feature sets. For a detailed explanation, please refer to "Section 3.2.1 - Weighted co-occurrence counting".
>
> **Q2: Performance w.r.t. the number of backbone model parameters**
>
> **A2:** Thank you for the suggestion. We conducted experiments to study the impact of backbone model size on performance. Specifically, we evaluated three model variants with varying parameter numbers:
>
> |**Variant**|**#Parameters**|**d_model**|**d_ff**|**num_layers**|**num_heads**|
> |-|-|-|-|-|-|
> |small|2.89M|64|256|2|2|
> |base|9.58M|128|1024|4|6|
> |large|23.35M|256|2048|4|6|
>
> We tested these variants on a small dataset (Sports) and a large dataset (CDs), with results summarized below:
>
> ||**Sports (N@10)**|**CDs (N@10)**|
> |-|-|-|
> |small|0.0261|0.0289|
> |base|**0.0264**|0.0416|
> |large|0.0242|**0.0451**|
>
> These results indicate that performance is influenced by both dataset size and model size (under a fixed number of tokens). On the smaller Sports dataset, the base model performs best, while the large model shows signs of overfitting. On the larger CDs dataset, the large model achieves the best performance, suggesting the dataset is sufficiently large to benefit from increased model size.
>
> **Q3: LLM-based sequential recommendation**
>
> **A3:** Thank you for the valuable suggestion. While our paper primarily focuses on action tokenization methods, LLM-based sequential recommendation is indeed closely related. Below is a high-level discussion of its relevance and connection to our work:
>
> When aligning LLMs with user preferences for sequential recommendation through instruction tuning on historical action sequences, the way these actions are tokenized plays a crucial role. We identify three main paradigms:
> 1. Text-based tokenization: Each action is represented as a textual string, which aligns naturally with LLMs' input modality. However, this approach leads to significantly long token sequences, resulting in both tokenization inefficiency and high inference latency.
> 2. Dense vector representations: Actions are represented as dense vectors, typically derived from pretrained semantic encoders or embedding tables. While this method is more efficient in terms of sequence length, it faces memory and scalability issues, especially since the number of items often exceeds the typical token vocabulary size of LLMs. Aligning LLMs with these continuous representations poses challenges in both engineering and optimization.
> 3. Discrete tokenization: Actions are tokenized into short sequences of discrete tokens drawn from a compact shared vocabulary (usually much smaller than that of typical LLMs). This strikes a balance between token length and memory efficiency, making it a practical solution for building LLM-based recommendation systems.
>
> We appreciate the reviewer's input and will include a more detailed discussion of LLM-based sequential recommendation in the final version, with proper citations to relevant literature.
>
> **Q4: Cross-dataset pretraining**
>
> **A4:** Thank you for highlighting this direction. While the transfer learning paradigm - pretraining on a diverse collection of datasets followed by fine-tuning on new datasets or platforms - has shown promising in retrieval-based recommendation methods (e.g., VQ-Rec [Hou et al., 2023]), it remains an open challenge for generative recommendation models.
>
> To the best of our knowledge, there is currently no existing work that successfully applies generative models with action tokenization to this pretraining - fine-tuning paradigm. Nevertheless, we agree this is an important and exciting research direction. We will clarify this point in the final version and plan to explore it further in future work.

---

> > ### Comment · Reviewer_DB8e · 2025-04-03
> >
> > Thank you for your response. I have increased the rating to 4.

---

> > > ### Author Response · Authors · 2025-04-03
> > >
> > > Thank you very much for reading our rebuttal and updating your rating! We sincerely appreciate your feedback and constructive comments, which have helped us improve our paper.

---

### Official Review · Reviewer_jhbj · 2025-03-14

**Overall Recommendation:** 3

**Summary:**

This paper introduces ActionPiece, a novel tokenization method for generative recommendation systems that incorporates context when tokenizing user actions. Unlike existing approaches (RQ-VAE, etc.) that tokenize each action independently, ActionPiece represents actions as unordered feature sets and builds a vocabulary by merging frequently co-occurring feature patterns both within individual actions and across adjacent actions. The authors also introduce set permutation regularization to handle the unordered nature of feature sets, enabling data augmentation during training and ensemble prediction during inference.

Experiments on three Amazon Review datasets demonstrate that ActionPiece consistently outperforms existing tokenization methods, improving NDCG@10 by 6.00% to 12.82%. Detailed analyses show that ActionPiece achieves significantly higher token utilization rates (up to 95.33%) and creates more efficient tokenized sequences. The authors validate their approach with thorough ablation studies.

**Claims And Evidence:**

* Tokenization is an important topic in RecSyS, esp. given recent focus on generative recommendations. Prior work has primarily studied either VQ/RQ-based quantization or directly utilizing raw ids.
	* (+) This paper proposes a new direction, tokenization in the (unordered) feature space, and validates that this results in significantly higher token utilization rate (56.9% -> 95.3%, Figure 5, Section 4.4.2) and better results esp when this tokenization strategy is combined with set permutation regularization (Table 4).
* Contextual tokenization is presented as a major contribution of this work, but the gains seem small from Table 3 (2.2).
	* (-) I also struggle to understand what exactly these contextual tokens look like for Amazon Review datasets; Section 4.5 doesn't quite help given I thought these datasets are text-/id-only. It would be valuable to present examples in the Appendix.

**Essential References Not Discussed:**

* N/A

**Experimental Designs Or Analyses:**

The experiment designs are generally thorough. Two issues:
* Figure 4: Why is the maximal vocabulary size limited to 40K? I would expect a point where the performance with large vocabularies starts to degrade due to overfitting.
* Table 3: It would be valuable to present results without either (3.1) or (3.2). This is because for sparse datasets like Sports or Beauty, SPR itself would be a valuable technique.

**Methods And Evaluation Criteria:**

Proposed methods:
* (+) Directly combining features into tokens (WordPiece/SentencePiece style) is an understudied problem in RecSys, and the authors have shown that this has benefits on some Amazon Review datasets.
* (-) I would like to see clearer examples illustrating how this work in practice (eg what the learned tokens look like).
* (-) How would ActionPiece scale to high cardinality id vocabularies (eg video ids), which is the most popular tokenization method in RecSys?

Evaluation Criteria:
* (+) The evaluation methods used, including normalized sequence length (NSL, Figure 4) to measure tokenization efficiency, token utilization rate to assess vocabulary usage (Figure 5), and comparison against both ID-based methods, RQ-VAE based methods, and other GR approaches on NDCG (Table 2) generally make sense. I appreciate the authors conducting thorough studies on NSL and utilization rate in particular.

**Other Comments Or Suggestions:**

N/A

**Other Strengths And Weaknesses:**

N/A

**Questions For Authors:**

* How effective is feature merging when features are of very high cardinality, which is typical for sparse id features (eg video ids)? This also may make some design choices impractical, eg "maintaining a hashtable to store co-occurrences of token pairs"
* When the inputs are all text, is ActionPiece fundamentally different from SentencePiece besides dropping the order constraints?  Could the authors provide examples of learned vocabularies for Sports/Beauty/CDs under ActionPiece for inspection?
* The way context information is incorporated in this paper is to merge features across adjacent "Actions", which seems to introduce a lot of complexity. How much value does this add?

**Relation To Broader Scientific Literature:**

* w.r.t. RecSys, studies of alternative tokenization methods besides RQ-VAE and raw ids is important given recent focus on sequential/generative recommendations.
* w.r.t. NLP, the method seems like a natural extension of SentencePiece to RecSys. However I'm confused why the authors would drop ordering within an "Action" and introduce ordering across "Action"s.

**Theoretical Claims:**

* I checked the time complexity analyses in the paper and they appear correct to me.

---

> ### Author Rebuttal · Authors · 2025-03-31
>
> We thank the reviewer for the valuable feedback! Below, we first provide a detailed example of learned vocabulary, followed by clarifications regarding the experiments and method design.
>
> **Q1: Example of learned vocabularies**
>
> **A1:** As detailed in Section E, we use vector-quantized (VQ) tokens as item features. We also add an extra token per item to avoid conflicts. Thus, each item is associated with a total of 5 features: 4 VQ tokens and one extra token. The union of these tokens forms the initial vocabulary of ActionPiece. Notably, we do not use raw item IDs or raw text tokens.
>
> To further clarify, we provide a concrete example from the Sports dataset. The item-to-feature mapping looks like:
>
> |**Item ID**|**Features**|
> |-|-|
> |B000BS0I2G|[170, 438, 519, 820, 1127]|
> |B000XHGE00|[163, 398, 564, 1023, 1068]|
> |...|...|
>
> The ActionPiece vocabulary is constructed by iteratively merging token pairs into new tokens. Each row in the table below represents a merge rule, sorted by the order in which the rules were learned:
>
> |**Source Tokens**|**Target Token**|
> |-|-|
> |(363, 763)|1154|
> |(269, 515)|1155|
> |...|...|
> |(465, 1202)|1204|
> |(369, 760)|1205|
> |...|...|
> |(241, 1040)|39999|
> |(30314, 39998)|40000|
>
> The final vocabulary consists of 1153 initial tokens (4×256 VQ tokens, 128 extra tokens, and 1 padding token) and merging rules.
>
> We promise to release our code and constructed vocabularies, allowing others to reproduce and extend our work.
>
> **Q2: Experiments with vocabulary size > 40k**
>
> **A2:** We agree that experimenting with larger vocabulary sizes would make our study more comprehensive. We conducted experiments on the Sports dataset:
>
> |**Vocab Size**|**N@10**|
> |-|-|
> |40k|0.0264|
> |60k|0.0260|
> |80k|0.0269|
> |100k|0.0266|
>
> As shown, increasing the vocabulary size does not consistently improve performance, suggesting that larger vocabularies may lead to overfitting.
>
> **Q3: More ablation study with SPR**
>
> **A3:** To further understand SPR's impact, we introduce two additional ablation variants on the Sports dataset by applying SPR to:
> 1. TIGER
> 2. Variant (2.1), which uses only the initial tokens (without merging).
>
> |**Method**|**N@10**|
> |-|-|
> |TIGER|0.0225|
> |TIGER + SPR|0.0202|
> |(2.1)|0.0215|
> |(2.1) + SPR|0.0205|
>
> As we can see, directly applying SPR leads to degraded performance in both cases. This suggests that SPR alone is not sufficient to improve generative models, regardless of whether the tokens are ordered or unordered.
>
> **Q4: Gains of contextual tokenization seem small**
>
> **A4:** Note that Sports and Beauty have relatively short sequence lengths (8.32 and 8.87 actions per sequence). In contrast, CDs has longer sequences, averaging 14.58 actions per sequence. Longer sequences offer more opportunities to leverage contextual information during tokenization. As expected, the performance gap between variant (2.2) and ActionPiece is more pronounced on CDs.
>
> **Q5: Handling high cardinality features like item IDs**
>
> **A5:** As mentioned in **A1**, we add one extra token per item, which allows us to uniquely index each item. This design eliminates the need to explicitly incorporate item IDs. Likewise, for other high-cardinality features, we can adopt a joint indexing mechanism as well, representing each feature using a combination of tokens from shared vocabularies.
>
> **Q6: Order within an action vs. across actions**
>
> **A6:** Item features such as title or price typically do not have an inherent ordering relative to one another. In sequential recommendation, the historical actions of a user are typically ordered by timestamp to capture behavioral dynamics. Therefore, we preserve and use the temporal order of actions in the sequence. Intuitively, while features within an action are unordered, the composition of those features across time can still reflect sequential patterns.
>
> **Q7: Comparison with text tokenization methods like SentencePiece**
>
> **A7:** While ActionPiece can be viewed as a variant of SentencePiece that relaxes the order constraints among features within each action, this relaxation is both non-trivial and beneficial. Modeling each action as an unordered set aligns better with the inherent structure of the data and leads to improved performance. To enable effective tokenization under this setup, we introduce techniques such as weighted counting and SPR. Ablation studies show that removing any of these components results in a performance drop.
>
> **Q8: Complexity of modeling contextual information**
>
> **A8:** We acknowledge that ActionPiece introduces additional complexity. This mirrors the evolution seen in language modeling: when subword methods like BPE were first introduced, they were also considered more complex than word- or character-level tokenization. Yet, over time, such methods proved to be significantly more effective and have become standard in modern NLP pipelines. Similarly, we argue that context-aware tokenization is a necessary step forward for effectively modeling action sequences.

---

> > ### Comment · Reviewer_jhbj · 2025-04-04
> >
> > Thanks authors for the responses. I have some further clarifying questions:
> >
> > a/ In your explanation for the Sports dataset, you said "The final vocabulary consists of 1153 initial tokens (4×256 VQ tokens, 128 extra tokens, and 1 padding token) and merging rules." But Sports dataset has 35,598 items. Why is the number of extra tokens 128 and not 35,598?
> >
> > b/ The given example reminds me of prior work on learned feature crossing (eg Deep Crossing: Web-Scale Modeling without Manually Crafted Combinatorial Features KDD'16, CAN: Feature Co-Action Network for Click-Through Rate Prediction WSDM'22). It might be useful to compare the proposed vocabulary merging algorithm with related work on this topic.

---

> > > ### Author Response · Authors · 2025-04-04
> > >
> > > **Re a:** Thank you for the follow-up question. The key idea is that the final token used to represent each item is **not a standalone item ID**, but rather a combination of multiple features that **together uniquely index an item**.
> > >
> > > Taking the example of item `B000BS0I2G`, it is represented by a 5-token sequence: `[170, 438, 519, 820, 1127]`. The first four tokens capture semantic features derived from the VQ process, and the fifth token is selected from the 128 extra tokens (indices 1025–1152) to distinguish between items that might otherwise share the same first four semantic tokens.
> > >
> > > The assumption is that no more than 128 items will share the same four-token semantic prefix (i.e., `[170, 438, 519, 820, xxx]`), which allows us to use one of the 128 extra tokens as a suffix to distinguish them. If a collision occurs (i.e., two items share the same four-token prefix), we assign different extra tokens from the 128-token pool to maintain uniqueness.
> > >
> > > This design ensures that all features work jointly to form a unique identifier for each item, rather than relying solely on the fifth token. It is inspired by similar practices such as TIGER [Rajput et al., 2023], which uses only 1024 tokens to represent all 35,598 items in the Sports dataset.
> > >
> > > ---
> > >
> > > **Re b:** Thank you for pointing out the connections to prior work on learned feature crossing. These are indeed relevant and valuable references.
> > >
> > > The key distinction is that such works primarily perform feature crossing at the model level, meaning that feature interactions are learned implicitly through network structures. In contrast, our method performs feature merging at the vocabulary level, enabling more efficient tokenization and modeling.
> > >
> > > While we did not directly compare to these specific models, we included several relevant baselines with similar design philosophies. For example, **HSTU** and our **variant (2.1)** in Table 3 use the same underlying item features as ActionPiece but provide them as flattened inputs - without merging - allowing the autoregressive model to learn feature interactions through its self-attention and feed-forward layers. This setup relies on model-level interaction learning, similar to the spirit of Deep Crossing and CAN.
> > >
> > > Our results show that ActionPiece, which performs vocabulary-level feature merging, outperforms these model-level baselines both in recommendation performance (Tables 2 & 3) and efficiency (Figure 4), especially in terms of normalized sequence length (NSL), where HSTU and variant (2.1) have NSL of 1, indicating significantly longer sequences.
> > >
> > > This observation parallels long-standing discussions in language modeling: byte-level models (akin to model-level feature merging) may sometimes achieve better perplexity/logloss, but are much less efficient due to longer token sequences. In contrast, token-level models (e.g., BPE, WordPiece) achieve better downstream performance and efficiency.
> > >
> > > We appreciate the reviewer's insightful comments and will incorporate these references and the discussion into the final version of the paper.
> > >
> > > ---
> > >
> > > Thank you once again for raising these discussions, which really have helped us improve the paper! We truly appreciate your time and engagement!

---

### Official Review · Reviewer_yBx7 · 2025-03-17

**Overall Recommendation:** 3

**Summary:**

this paper proposed ActionPiece, a tokenization strategy for generative recommendation systems. the main idea of ActionPiece can be summarized as following: after collecting all features of each action set, the authors proposed to reconstruct the user historical action sequences by i) vocabulary construction: use simple counting to compute the co-occurrence of existing tokens and update token pairs based on the co-occurrence. ii) segmentation: generate random permutation and apply BPE. A transformer encoder-decoder model is trained on top of the proposed tokenization technique and experiments on 3 amazon recommendation datasets were reported to show that the proposed tokenization technique outperforms existing baselines.

**Claims And Evidence:**

Yes.

**Essential References Not Discussed:**

None.

**Experimental Designs Or Analyses:**

yes

**Methods And Evaluation Criteria:**

Yes.

**Other Comments Or Suggestions:**

None

**Other Strengths And Weaknesses:**

Strengths:
- the proposed tokenization technique provides great intuition how to restructuring the user historical sequence by combining the tokenization ideas from LLM.
- the proposed method is based on counting, merging and BPE, which are simple and easy to understand.
- the proposed method achieves the best performance on several public datasets.


Weaknesses：
- the experiments were conducted on small datasets. no results on industrial scale recommendation systems were reported. this makes the ideas of the paper weaker since it hasn't been tested in real world.
- no theoretical justifications.

**Questions For Authors:**

- can the authors comment more on "Set permutation regularization"? specifically, why random permutation of each action set is critical to the performance improvement.

**Relation To Broader Scientific Literature:**

None

**Theoretical Claims:**

No theoretical results were provided.

---

> ### Author Rebuttal · Authors · 2025-03-31
>
> We sincerely thank the reviewer for the thoughtful and constructive feedback. We appreciate your recognition of the intuition, simplicity, and effectiveness of ActionPiece tokenization technique.
>
> **Q1: On experiments with industrial-scale datasets or online A/B testing**
>
> **A1:** We acknowledge the reviewer's concern regarding the absence of industrial-scale experiments. Due to resource constraints, we were unable to run experiments on industrial-scale offline datasets or perform online A/B tests. However, to address the scalability concern, we included the CDs dataset in our evaluation, which contains over 1 million user-item interactions. To our knowledge, this dataset is among the largest publicly available datasets used for studying generative recommendation or action tokenization methods.
>
> **Q2: On set permutation regularization (SPR)**
>
> **A2:** SPR benefits the model from multiple perspectives:
>
> * *Token utilization perspective:*
>
>     SPR effectively prevents the features of a single action from being consistently merged into the most compressed (high-level) tokens. Instead, it allows the action to be tokenized into both high-level and low-level tokens, depending on the permutation and token merging rules. This increases the number of tokens actively involved during both training and inference. As shown in Figure 5 and discussed in Section 4.4.2, SPR significantly improves token utilization - from 56.89% to 95.33% by the 5th epoch - indicating that a greater proportion of tokens are trained after applying SPR.
>
> * *Data augmentation perspective:*
>
>     From the perspective of data augmentation, SPR enriches the token sequences available for model training. Without SPR, each action sequence can only be tokenized into a single, fixed token sequence. In contrast, SPR allows each action sequence to be tokenized in multiple ways (as shown in Figure 1). While these augmented sequences preserve the same semantic information, they expose the model to richer token patterns. Training on these diverse token sequences helps the model generalize better, as evidenced by the performance of variant (3.1) in Table 3.
>
> * *Ensemble perspective:*
>
>     SPR also enables inference-time augmentation. A given input action sequence can be augmented into multiple token sequences during inference. Each sequence may yield a different ranking of the next possible items. By ensembling these recommendation results, overall performance can be enhanced, as demonstrated by variant (3.2) in Table 3 and further illustrated in Figure 6.
>
> We thank the reviewer again for their helpful comments. We will incorporate the above clarifications and discussions in the final version of the paper.

---

### Official Review · Reviewer_w42X · 2025-03-18

**Overall Recommendation:** 3

**Summary:**

This paper introduces ​ActionPiece, a context-aware action sequence tokenization method for Generative Recommendation (GR). The main contributions are as follows: (1)​Context-aware tokenization, which represents user action sequences as sequences of unordered feature sets and then merges frequently co-occurring feature patterns (both intra-action and cross-action) to build a vocabulary capturing contextual dependencies, enabling distinct tokens for the same action in different contexts. (2) ​Set Permutation Regularization (SPR), leveraging the unordered nature of feature sets by generating multiple semantically equivalent token sequences through random permutations, serving as data augmentation during training and ensemble sources during inference to enhance generalization.
Experimental results demonstrate ActionPiece’s performance over existing methods (ID-based, feature-enhanced, and GR baselines) on Amazon datasets (Sports,Beauty,CDs), achieving NDCG@10 improvements of 6.00%–12.82%. Ablation studies confirm the necessity of context awareness, weighted co-occurrence counting, and SPR, with the latter boosting token utilization from 56.89% to 87.01%. Inference-time ensemble over 5 permutations balances performance and computational cost. The work pioneers context-aware tokenization in recommendation systems, enabling finer-grained semantic modeling for GR.

**Claims And Evidence:**

This paper demonstrates stable improvements of the proposed method across three datasets compared to ID-based, feature-enhanced, and generative baselines. The results align with the claim that context-aware tokenization improves performance. The inclusion of ablation studies further validates the necessity of key components like context-aware merging and SPR. However, some baseline results from Sports and Beauty datasets are directly taken from original papers, while the results in CDs dataset are reimplemented, which may result in inconsistency.

**Essential References Not Discussed:**

This paper comprehensively cites relevant works that provide necessary context for its key contributions.

**Experimental Designs Or Analyses:**

I have thoroughly reviewed the experimental section of this paper. Overall, the experimental design is largely reasonable, but certain limitations still exist. In the main experiments, the proposed method achieves state-of-the-art performance; however, there is inconsistency in baseline results across different datasets as has been mentioned before. The ablation study validates the effectiveness and scalability of each component of the method. However, TIGER achieves optimal results under a vocabulary size of 4×2^8, which suggests that smaller vocabulary sizes might yield better performance, which may result in unfairness in the main experimental comparisons. Additional experiments analyze the method’s performance under varying parameters, and their design and conclusions are generally well-founded.

**Methods And Evaluation Criteria:**

This paper introduces ActionPiece, a context-aware tokenization method for generative recommendation (GR) systems. Overall, the proposed method is effective but has certain limitations:
1. Contextual Action Sequence Tokenization: Representing actions as unordered feature sets and merging co-occurring features (within or across adjacent actions) into context-sensitive tokens during vocabulary construction. While this method incorporates contextual information, it fails to model inherently ordered features (e.g., Cosmetics → Lip Products → Lipstick), which naturally follow hierarchical dependencies.
2. Set Permutation Regularization (SPR): Generating multiple semantically equivalent token sequences by permuting features within sets, enhancing training through data augmentation and inference via ensemble predictions. Although experimentally proven effective, SPR may introduce additional computational overhead, impacting overall system efficiency.
3. Efficient Implementation: Using linked lists and lazy-update heaps to optimize vocabulary construction, accelerating algorithmic execution.
The selected benchmarks are reasonable but lack diversity. While Amazon Benchmarks (Sports, Beauty, CDs) are standard datasets in recommendation research, this paper does not evaluate performance on datasets beyond the Amazon domain.

**Other Comments Or Suggestions:**

1.There is an issue with line numbering in Algorithm 2 in the appendix.
2.The first letter in Figure 3 should be capitalized.

**Other Strengths And Weaknesses:**

1.The details of the inference process need to be further explained.

**Questions For Authors:**

1.How does set permutation affect training and inference efficiency?
2.Why are some papers (e.g. IDGenrec in [Tan et al., 2024;] and LETTER in [Wang et al., 2024a] ) with better performance cited but not included in the baseline comparison?

**Relation To Broader Scientific Literature:**

None

**Theoretical Claims:**

The theoretical claims in this paper primarily focus on  weighted co-occurrence counting  and  time complexity calculation, with the logic being rigorously structured and the derivations mathematically sound.

---

> ### Author Rebuttal · Authors · 2025-03-31
>
> Thank you very much for your time and thoughtful feedback. We greatly appreciate your constructive and insightful suggestions. Below, we address your concerns regarding the experiments, followed by additional discussions.
>
> **Q1: Inconsistent experimental settings**
>
> **A1:** We included results on the CDs dataset to demonstrate performance of compared methods on a large-scale dataset. However, to the best of our knowledge, there are no publicly available results of generative recommendation methods on CDs. Therefore, we carefully followed the same experimental settings used in public benchmarks such as Sports and Beauty to ensure fair comparisons.
>
> We would like to clarify that the "CDs" dataset used in LC-Rec [Zheng et al., 2024] is from a different version of the Amazon Reviews dataset. Specifically, our work uses the Amazon 2014 version, whereas LC-Rec uses a 2018 version.
>
> **Q2: Results on datasets beyond Amazon**
>
> **A2:** Thank you for the suggestion to improve the comprehensiveness of our experimental evaluation. We additionally conducted experiments on another widely used public benchmark Yelp, following the experimental setting in the LETTER [Wang et al., 2024a] paper.
>
> |**Yelp**|**N@10**|
> |-|-|
> |TIGER|0.0213|
> |SPM-SID|0.0226|
> |LETTER|0.0231|
> |ActionPiece|**0.0255**|
>
> These results demonstrate that ActionPiece achieves better performance than the compared baselines on Yelp as well.
>
> **Q3: Results of TIGER with smaller vocabularies**
>
> **A3:** In our original submission, we compared ActionPiece to both (1) TIGER with larger vocabularies and (2) SPM-SID, aiming to ensure a fair comparison in terms of vocabulary size and to demonstrate that the improvements are not solely due to having more tokens.
>
> We appreciate the reviewer's insightful suggestion and conduct additional experiments using TIGER variants with smaller vocabulary sizes:
>
> |**Sports**|**N@10**|
> |-|-|
> |TIGER (4×48)|0.0231|
> |TIGER (3×256)|0.0220|
> |TIGER (4x256, original)|0.0225|
> |ActionPiece|**0.0264**|
>
> While one TIGER variant with smaller vocabularies can perform slightly better than the numbers reported in the original TIGER paper, ActionPiece still achieves significantly better performance than all TIGER variants.
>
> **Q4: Baselines like IDGenRec and LETTER were not compared**
>
> **A4:** Each generative recommendation baseline in our paper was selected to represent one different tokenization paradigm, consistent with those in Table 1. Our core contribution is to introduce *context-aware* tokenization as a novel and promising paradigm, rather than to claim that ActionPiece is the best action tokenization method.
>
> That said, we agree that additional comparisons are helpful. Below are the results comparing ActionPiece with IDGenRec (after fixing the data leakage issue in https://github.com/agiresearch/IDGenRec/issues/1):
>
> ||**Sports (N@10)**|**Beauty (N@10)**|
> |-|-|-|
> |IDGenRec|0.0223|0.0404|
> |ActionPiece|**0.0264**|**0.0424**|
>
> The comparison with LETTER has been included in our response to **Q2** above.
>
> **Q5: Tokenize inherently ordered features**
>
> **A5:** Thank you for highlighting this important direction. Injecting hierarchical structure into the tokenization process is indeed a challenging task for most existing action tokenization methods, especially those that rely on quantization techniques. While text-based tokenization can naturally capture such hierarchies, it suffers from tokenization inefficiencies. We will discuss these limitations and the trade-offs in the final version of the paper.
>
> **Q6: Efficiency of set permutation regularization (SPR)**
>
> **A6:** In terms of training, the efficiency is comparable to existing methods. The feature permutation operations are performed on the CPU and run asynchronously alongside GPU-based model updates.
>
> For inference, while SPR introduces additional computational overhead in terms of FLOPs, the latency remains comparable. This is because the augmented versions of each test case can be processed in parallel across multiple GPUs, resulting in inference latency that is comparable to the non-augmented single-GPU setting.
>
> **Q7: Details of the inference process**
>
> **A7:** Our model inference process follows TIGER. The decoder autoregressively generates token sequences for the target items. During training, we use the original item features as labels without any augmentation or token merging. At inference time, we apply beam search to generate the top-ranked token sequences. The most probable token sequences (in other words, prefixes) are retained in the current beam (with beam size detailed in Table 7), and the model continues generating tokens one at a time until the desired generation length is reached.
>
> **Q8: Formatting issues in Algorithm 2 and Figure 3**
>
> **A8:** Thank you for your careful and detailed review. We appreciate your effort in catching these formatting issues and will address them in the final version.

---

### Decision · Program_Chairs · 2025-05-01

**Decision:**

Accept (spotlight poster)

**Comment:**

The authors provide novel contributions to the nascent but fast growing field of Generative Recommendation models.  In response to reviewer concerns, the authors provided a solid and convincing rebuttal and hence there is general agreement from the reviewers that the work should be accepted.

Reviewers would like to see the additional discussion and results discussed in the rebuttal incorporated into the paper Appendices and referenced in the main text where relevant.

Two particular points stood out in post-rebuttal discussion that I would like to highlight as the authors make final revisions:

1. **The motivation and need for context-awareness**:  The ablation studies in Table 3 appear to show little degradation when context-awareness is removed, yet the paper appears to strongly argue for its importance.  The authors are requested to revisit their motivation and arguments in light of their experimental results to ensure they are correctly motivating and highlighting the key components that are empirically validated to offer the largest gains.


   To this broader point of framing the motivation and argumentation correctly, one reviewer comments: "even ignoring 'contextual tokenization', the authors are more or less learning *explicit feature crosses* with their proposed token merging algorithm, which should be a strict improvement over prior work that constructs codebook solely based on RQ-VAE".

2. **Concerns about reproducibility**:  One reviewer comments that reproduction of the proposed token merging logic would take a lot of work.  No code was made available with the paper submission, but I strongly encourage the authors to make an effort to allow reproduction of their work for the sake of future papers that would like to extend this paper and compare to its results.